# PROVABLY SAFE REPRESENTATION LEARNING IN CMDPS: A PRIMAL-DUAL APPROACH

## ABSTRACT

We study representation learning in low-rank Constrained Markov Decision Processes (CMDPs) with unknown dynamics, where the agent must maximize rewards under safety constraints. While representation learning has significantly advanced for unconstrained MDPs, its extension to CMDPs remains open due to the critical challenge of safe exploration under learned features, particularly concerning the management of soft constraint violation. In this work, we propose REP-PD, the first algorithm that provably integrates representation learning with policy optimization in low-rank CMDPs. By iteratively learning a low-rank transition representation via MLE and utilizing a composite Q-function tied to the unconstrained Lagrangian, REP-PD guides policy updates to balance reward maximization, exploration, and robust constraint adherence. Through this approach, REP-PD achieves a near-optimal policy with a sampling complexity bound independent of the state space dimension without prior feature knowledge. Notably, REP-PD's regret matches the lower bounds for unconstrained low-rank MDPs, achieving strong performance concerning soft constraint violation. We then consider a stronger hard constraint violation metric, where the agent must strictly satisfy constraints at all times, and propose REP-PD-hard by designing a novel policy optimization module. Our work thus provides a robust and theoretically grounded approach to representation learning in constrained reinforcement learning, with guarantees on bounded soft and hard constraint violation.

## 1 INTRODUCTION

Constrained Markov Decision Processes (CMDPs) provide a principled framework for constrained reinforcement learning (RL), where agents must maximize rewards while satisfying constraints. This framework is particularly relevant for safety-critical domains where constraints encode safety requirements (1; 2). Several remarkable algorithms have been developed within this framework (3; 4; 5; 6; 7).

On the theoretical front, provably constrained RL has also garnered increasing attention. Recent works (8; 9; 10; 11; 12; 13) primarily focus on tabular setup, achieving sample complexity dependent on the cardinality of the state space, rendering them inapplicable to CMDPs with continuous state spaces. Indeed, solving CMDPs in continuous or high-dimensional state spaces poses a significant challenge due to the curse of dimensionality, which complicates both policy optimization and exploration. Linear CMDP methods (14; 15; 16; 17; 18; 19) partially mitigate this issue by assuming transitions are linearly representable via low-dimensional latent features, achieving dimension-independent sample complexity. However, they crucially require prior knowledge of the latent features, a restrictive assumption in practice as handcrafting accurate representations for complex dynamics is often infeasible, leaving the challenge of solving CMDPs in unknown, complex environments largely unaddressed.

Representation learning has proven effective in combating the curse of dimensionality for unconstrained MDPs, enabling agents to discover latent features without manual engineering (20; 21; 22; 23; 24; 25; 26). However, extending these techniques to CMDPs presents difficulties. It is hard to balance exploration, reward maximization, and constraint satisfaction under learned features. The fundamental challenge lies in performing safe and effective exploration guided solely by learned features in the presence of constraints. The agent must explore to learn both the dynamics and the

constraint within the feature representation. The uncertainty of features makes it extremely difficult to design exploration strategies that are both sufficiently informative for representation learning and robustly safe with respect to the constraints. Consequently, representation learning for CMDPs in unknown environments remains an open problem.

In this paper, we propose REP-PD, the first provable algorithm for low-rank CMDPs that *jointly* learns latent representations and optimizes policies while ensuring soft constraint violation satisfaction. REP-PD considers the unconstrained Lagrangian function of the original constrained problem and guides the policy to explore with a composite Q-function associated with the Lagrangian, ensuring both reward maximization and robust constraint adherence. Specifically, we construct a composite Q-function, which merges the reward and cost utility Q functions. Besides, an exploration bonus is integrated into the composite Q-function to facilitate exploration. The policy is updated by maximizing this composite Q-function. In each iteration of REP-PD, we first extend the empirical datasets using a strategically designed exploration policy that mixes the current policy with uniform exploration. Then, a low-rank representation of the transition function is learned using MLE from collected empirical data, and the exploration bonus is updated based on the learned representation. Subsequently, leveraging both the learned representation and this exploration bonus, we update the composite Q-function and the policy accordingly. Our approach achieves a near-optimal policy with a sampling complexity bound independent of the state space dimension. Notably, REP-PD's regret matches the lower bounds for unconstrained low-rank MDPs (21), indicating that our constraint-aware extensions do not introduce excessive overhead. We further extend our algorithm to address the more stringent metric of hard constraint violation by designing a novel policy optimization module. This algorithm, REP-PD-hard, is then shown to have a theoretical bound on its performance under this stricter metric. Our work thus provides a robust and theoretically grounded approach to representation learning in constrained reinforcement learning, with guarantees on bounded soft and hard constraint violation. Specifically, we make the following contributions:

- We propose REP-PD, the first provably efficient algorithm for low-rank CMDPs that jointly learns latent transition representations and optimizes policies under constraints. By interleaving MLE-based representation learning with Lagrangian multiplier updates, REP-PD adaptively balances exploration, reward maximization, and constraint adherence in high-dimensional state spaces without prior knowledge of transition features.

- We establish the theoretical guarantees for REP-PD, demonstrating its near-optimal performance. Specifically, REP-PD achieves near-optimal regret and soft constraint violation bounds of $\tilde{\mathcal{O}}(|\mathcal{A}|d^2\sqrt{K})$. These results scale polynomially with the feature dimension $d$ rather than the state space dimension, enabling efficient learning in high-dimensional settings. Our novel analysis links learned feature error to constraint violation probabilities, avoiding pointwise transition model error bounds typically required in prior linear CMDP works (14; 15; 16; 17; 18). Moreover, we propose a variant of our algorithm, REP-PD-0, which is proven to achieve zero soft constraint violation for sufficiently large $K$.

- We propose REP-PD-hard to tackle the more stringent metric of *hard* constraint violation. REP-PD-hard incorporates a novel policy optimization module to achieve $\tilde{O}\left(\frac{|\mathcal{A}|d^2K^{3/4}}{(1-\gamma)^2}\right)$ regret and $\tilde{O}\left(\frac{|\mathcal{A}|d^2\sqrt{K}}{(1-\gamma)^2}\right)$ hard constraint violation bounds. Notably, the $\tilde{O}(\sqrt{K})$ hard constraint violation bound aligns with the optimal bounds for tabular and linear settings.

## 1.1 RELATED WORKS

**Provably Constrained RL**   CMDPs have emerged as a fundamental framework for safe RL, requiring agents to maximize rewards while satisfying cumulative cost constraints (2; 27). Several remarkable works have been developed on this foundation (3; 4; 5; 6; 7; 28). Meanwhile, there has been a growing body of work on provably safe reinforcement learning. Many works (8; 9; 10; 11; 12; 13; 29; 30) considered tabular set-up. Among the best known regret and constraint violations achieved are $\tilde{\mathcal{O}}(\sqrt{|\mathcal{S}|^2|\mathcal{A}|H^6K})$, which can not cope with the large state space.

Recent advances in linear CMDPs (31; 14; 32; 19; 15; 16; 17; 18) achieve dimension-independent sample complexity by assuming transitions are linearly representable via low-dimensional features $\phi^*(s,a)$. These methods leverage optimistic exploration bonuses tied to $\phi^\star(s,a)$, ensuring safety via confidence intervals on transition dynamics. However, their reliance on *predefined* latent fea-

| MDP | Paper | Known Env | Regret | Violation |
|---|---|---|---|---|
| Tabular | (8) | Yes | $\tilde{O}\left(\sqrt{|\mathcal{S}|^3|\mathcal{A}|H^4K}\right)$ | $\tilde{O}\left(\sqrt{|\mathcal{S}|^3|\mathcal{A}|H^4K}\right)$ |
| | (12) | Yes | $\tilde{O}\left(\sqrt{|\mathcal{S}|^3|\mathcal{A}|H^6K}\right)$ | $0$ |
| Linear | (19) | Yes | $\tilde{O}\left(\sqrt{d^2H^6K}\right)$ | $\tilde{O}\left(\sqrt{d^2H^6K}\right)$ |
| | (15) | Yes | $\tilde{O}\left(\sqrt{d^3H^3K}\right)$ | $\tilde{O}\left(\sqrt{d^3H^3K}\right)$ |
| | (15) | Yes | $\tilde{O}\left(\sqrt{d^3H^5K}\right)$ | $0$ |
| Low-rank | **REP-PD** | **No** | $\tilde{O}\left(|\mathcal{A}|\sqrt{\frac{d^4K}{(1-\gamma)^4}}\right)$ | $\tilde{O}\left(|\mathcal{A}|\sqrt{\frac{d^4K}{(1-\gamma)^4}}\right)$ |
| | **REP-PD-0** | **No** | $\tilde{O}\left(|\mathcal{A}|\sqrt{\frac{d^4K}{(1-\gamma)^6}}\right)$ | $0$ |

Table 1: Comparison of tabular/linear/low-rank CMDPs with bounded *soft* constraint violation results. $|\mathcal{S}|$, $|\mathcal{A}|$, $d$, $H$, $\gamma$ and $K$ denote state space size, action space size, feature dimension, horizon, discounted factor, and number of episodes, respectively.

| MDP | Paper | Known Env | Known Safe Action Set | Regret | Violation |
|---|---|---|---|---|---|
| Tabular | (8) | Yes | Yes | $\tilde{O}\left(\sqrt{|\mathcal{S}|^3|\mathcal{A}|H^4K}\right)$ | $\tilde{O}\left(\sqrt{|\mathcal{S}|^3|\mathcal{A}|H^4K}\right)$ |
| | (13) | Yes | Yes | $\tilde{O}\left(\sqrt{|\mathcal{S}|^2|\mathcal{A}|H^6K}\right)$ | $0$ |
| Linear | (14) | Yes | Yes | $\tilde{O}\left(\sqrt{d^3H^4K}\right)$ | $0$ |
| | (16) | Yes | No | $\tilde{O}\left(\sqrt{d^3H^4K}\right)$ | $\tilde{O}\left(\sqrt{d^3H^4K}\right)$ |
| | (17) | Yes | No | $\tilde{O}\left(\sqrt{d^3H^4K}\right)$ | $\tilde{O}\left(\sqrt{dH^2K}\right)$ |
| | (18) | Yes | Yes | $\tilde{O}\left(\sqrt{d^5H^8K}\right)$ | $0$ |
| Low-rank | **REP-PD-hard** | **No** | **No** | $\tilde{O}\left(\frac{|\mathcal{A}|d^2K^{3/4}}{(1-\gamma)^2}\right)$ | $\tilde{O}\left(\frac{|\mathcal{A}|d^2\sqrt{K}}{(1-\gamma)^2}\right)$ |

Table 2: Comparison of tabular/linear/low-rank CMDPs with bounded *hard* constraint violation results. $|\mathcal{S}|$, $|\mathcal{A}|$, $d$, $H$, $\gamma$ and $K$ denote state space size, action space size, feature dimension, horizon, discounted factor, and number of episodes, respectively.

tures—assumed to be known a priori or handcrafted—limits applicability to complex scenarios where transition structures are unknown and must be learned from data. In contrast to these works, our study targets CMDPs with *unknown* latent features, where the low-dimensional transition structure must be learned directly from data. While prior art on linear CMDPs focuses on exploitation of known features, we introduce a data-driven framework that jointly learns latent features $\hat{\phi}$ via MLE, eliminating the need for prior knowledge of $\phi^*$. To our knowledge, the only prior work on low-rank CMDPs is that of (33), which focused on reward-free RL in a finite-horizon setting. In contrast, our work addresses the reward-known problem, and our method is applicable to the infinite-horizon regime.

**Representation Learning in RL** A growing body of research has focused on theoretically grounded representation learning in RL, aiming to derive low-dimensional features that efficiently capture environment dynamics. For example, works such as (34; 35; 36) investigated representation learning in block MDPs, leveraging hierarchical or modular state decompositions to improve sample efficiency. (37) studied representations under a Gaussian noise model, establishing guarantees for feature learning in noisy environments.

Low-Rank MDPs have emerged as a pivotal framework for representation learning in structured environments. Model-based approaches, including (20; 21; 38; 22; 23; 24; 39) learn latent transition features by leveraging model classes of transition probabilities, achieving sample-efficient exploration under realizability assumptions. In contrast, model-free methods like (25) eschew explicit transition models and proven efficiency under minimal reachability conditions. (40) developed reward-free exploration in low-rank MDPs. Recent extensions to multi-agent settings, such as (26; 41; 42), further explored representation learning in Markov games. Notably, none of these works address safety constraints in CMDPs. While existing methods achieve dimension-agnostic

sample complexity in reward-driven settings, they lack mechanisms to ensure constraint satisfaction during exploration or policy optimization. To our knowledge, this work presents the first representation learning framework tailored for reward known low-rank CMDPs, bridging the gap between unsupervised feature discovery and provable safety guarantees.

## 2 BACKGROUND

We consider an episodic constrained MDP, denoted by $(\mathcal{S}, \mathcal{A}, P^*, r, g, \gamma, d_1)$ where $\mathcal{S}$ is the state space, $\mathcal{A}$ is the action space, $P^*$ is the transition function, $r$ is the reward function, $g$ is the utility function, $\gamma$ is the discounted factor, and $d_1$ is the initial distribution. Following prior work (43; 44; 21), we assume trajectory reward is normalized, i.e., for any trajectory $\{s_h, a_h\}_{h=1}^\infty$, we have $\sum_{h=1}^\infty \gamma^h r(s_h, a_h) \in [0, 1]$ and $\sum_{h=1}^\infty \gamma^h g(s_h, a_h) \in [0, 1]$.

In an infinite-horizon discounted CMDP, each episode starts with state $s_1 \sim d_1$. Then at each step $h$, the agent observes state $s_h \in \mathcal{S}$, picks an action $a_h \in \mathcal{A}$, and receives a reward $r(s_h, a_h)$ and a utility $g(s_h, a_h)$. The MDP then transitions to $s_{h+1}$ which is drawn from $P^*(\cdot|s_h, a_h)$. In this paper, we assume access to reward and utility function feedback, but does not require access to the transition model. For any $s \in \mathcal{S}$, $\pi(a|s)$ denotes the probability of selecting action $a \in \mathcal{A}$ when the state is $s$ under policy $\pi$. We define the expected value function for the discounted reward under policy $\pi$ starting from state $s$ as:

$$V_{P^*,r}^\pi(s) = \mathbb{E}_{P^*,\pi}\left[\sum_{h=1}^\infty \gamma^h r(s_h, a_h)|s_1 = s\right], \tag{1}$$

where $\mathbb{E}$ is taken with respect to the policy $\pi$ and the transition function $P^*$. Similarly, we define the action-value function for the reward as:

$$Q_{P^*,r}^\pi(s, a) = \mathbb{E}_{P^*,\pi}\left[\sum_{h=1}^\infty \gamma^h r(s_h, a_h)|s_1 = s, a_1 = a\right]. \tag{2}$$

For simplicity, when the context is clear, we sometimes abbreviate $V_{P^*,r}^\pi$ and $Q_{P^*,r}^\pi$ as $V_r^\pi$ and $Q_r^\pi$, respectively. We also define the value function for the utility $V_g^\pi(s)$, and the action-value function for the utility $Q_g^\pi(s, a)$ analogously. We denote $V_j^\pi(s)$, and $Q_j^\pi(s, a)$ for $j = r, g$.

**Definition 1.** *For brevity, we denote $P^* V_j^\pi(s, a) = \mathbb{E}_{s' \sim P^*(\cdot|s,a)} V_j^\pi(s')$ for $j = r, g$.*

Using this notation, the Bellman's equation associated with the policy $\pi$ becomes

$$Q_j^\pi(s, a) = (r + \gamma P^* V_j^\pi)(s, a) \tag{3}$$

Note that $V_j^\pi(s) = \langle \pi(\cdot|s), Q_j^\pi(s, \cdot) \rangle_\mathcal{A}$, where $\langle \pi(\cdot|s), Q_j^\pi(s, \cdot) \rangle_\mathcal{A} = \sum_{a \in \mathcal{A}} \pi_h(a|s) Q_j^\pi(s, a)$.

The objective of the learning agent is to find an optimal policy that maximizes the expected reward while satisfying a safety constraint on the expected utility:

$$\max_\pi V_r^\pi(s_1), \qquad \text{subject to } V_g^\pi(s_1) \geqslant b. \tag{4}$$

where $b \in (0, \frac{1}{1-\gamma}]$ is a given threshold. While we focus on a single constraint for simplicity, our method naturally extends to handle multiple constraints. We denote the optimal policy as $\pi^*$ which solves the above optimization problem.

In the absence of prior knowledge about the constraint, the agent must balance exploration and exploitation while ensuring that the policy satisfies the constraint. To this end, we allow the policy to occasionally violate the constraint and aim to minimize the regret and the total constraint violations over $K$ episodes. This approach is consistent with existing literature on constrained RL (8; 19; 45).

To formally define the performance metrics, Let $\pi_k$ denote the policy employed by the agent in episode $k$. The regret and constraint violation are defined as follows.

$$\text{Regret}(K) = \sum_{k=1}^K V_r^{\pi^*}(s_1) - V_r^{\pi_k}(s_1), \tag{5}$$

$$\text{Violation}_{soft}(K) = \left[\sum_{k=1}^K (b - V_g^{\pi_k}(s_1))\right]_+, \quad \text{Violation}_{hard}(K) = \sum_{k=1}^K \left[b - V_g^{\pi_k}(s_1)\right]_+,$$

where $[z]_+ = \max\{z, 0\}$. The regret measures the gap between the total reward value by following the optimal policy $\pi^*$, and the total reward value obtained by following agent's policy $\pi_k$ at episode $k$ over $K$ episodes. The soft constraint violation quantifies the cumulative difference between the desired constraint threshold and the achieved utility, allowing for both positive and negative deviations over time. In contrast, the hard constraint violation is a stricter metric that measures only the cumulative positive deviations. As shown in (16), even zero soft violation may lead to hard constraint violation that grows linearly with the number of episodes, while soft constraint violation can only grow sub-linearly with the number of episodes if the hard constraint violation grows sub-linearly.

**Dual problem and Slater's Condition.** We then introduce some additional notation that will be used throughout this paper.

**Definition 2.** *For any policy $\pi$ and Lagrangian multiplier $Y$, we define the composite value function and Q-function as $V^{\pi,Y}(s) = V_r^\pi(s) + Y V_g^\pi(s)$ and $Q^{\pi,Y}(s,a) = Q_r^\pi(s,a) + Y Q_g^\pi(s,a)$.*

We can cast problem (4) as a saddle point problem $\max_\pi \min_Y \mathcal{L}(\pi, Y)$, with Lagrangian $\mathcal{L}(\pi, Y) = V_r^\pi(s_1) + Y(V_g^\pi(s_1) - b) = V^{\pi,Y} - Yb$. Although the Lagrangian is non-concave in $\pi$ (46), the strong duality holds (47). Hence, there exists an optimal Lagrangian multiplier $Y^*$ such that $\max_\pi \mathcal{L}(\pi, Y^*)$ recovers the optimal reward value function.

We assume the following Slater's condition in this paper.

**Assumption 1** (Slater's Condition). *There exists $\theta > 0$, and a policy $\bar\pi$, such that $V_g^{\bar\pi}(s_1) \geqslant b + \theta$,*

The Slater's condition is mild in practice and widely used in prior work (19; 8; 48). We use the properties of the Slater's condition to bound the performance of our algorithm.

Under the Slater's condition, $Y^\star$ is bounded.

**Lemma 3** (Boundedness of $Y^*$). *The optimal dual-variable $Y^* \leqslant \dfrac{V_r^{\pi^*}(s_1) - V_r^{\bar\pi}(s_1)}{\theta} \leqslant \dfrac{1}{\theta}$.*

We set $\xi = 2/\theta$ in the following analysis.

**Low-rank CMDP.** In this paper, we study low-rank CMDPs, which are characterized by a low-rank decomposition of the transition function. Specifically, we assume the following:

**Assumption 2.** *[low-rank CMDP] The transition function of the CMDP $P^* : \mathcal{S} \times \mathcal{A} \to \Delta(\mathcal{S})$ admits a low rank decomposition with rank $d \in \mathbb{N}$ if there exists two embedding functions $\phi^* : \mathcal{S} \times \mathcal{A} \to \mathbb{R}^d, \mu^* : \mathcal{S} \to \mathbb{R}^d$ such that*

$$\forall s, s' \in \mathcal{S}, a \in \mathcal{A} : P^*(s'|s,a) = \langle \phi^*(s,a), \mu^*(s') \rangle. \tag{6}$$

*For normalization, we assume $\|\phi^*(s,a)\| \leqslant 1$ for all $(s,a)$ and for any functions $g : \mathcal{S} \to [0,1]$, $\|\int \mu^*(s)g(s)\,ds\|_2 \leqslant \sqrt{d}$.*

**Remark 4.** *The low-rank CMDP assumption, characterized by the existence of low-dimensional embedding functions $\phi^*(s,a)$ and $\mu^*(s')$, is widely adopted structural assumption in RL (20; 21; 23). This assumption encompasses several common settings as special cases, including tabular CMDPs, linear CMDPs, and block CMDPs.*

**Function approximation setup and MLE oracle.** Since the transition function $P^*$ is unknown, we employ a function class $\mathcal{M} = \{(\mu, \phi) : \mu \in \Theta, \phi \in \Gamma\}$ to characterize $\mu^\star$ and $\phi^\star$, and by extension, to approximate $P^*$. To select the appropriate transition $\hat{P}$ from the function class $\mathcal{M}$, we assume access to a supervised learning-style Maximum Likelihood Estimation (MLE) oracle.

**Definition 5** (MLE Oracle). *Given a model class $\mathcal{M}$ and a dataset $\mathcal{D}$ in the form of $(s, a, s')$, the MLE oracle returns the maximum likelihood estimator $\hat{P} = (\hat{\mu}, \hat{\phi}) = \arg\max_{(\mu,\phi)\in\mathcal{M}} \mathbb{E}_\mathcal{D} \ln(\mu(s')^\top \phi(s,a))$.*

## 3 REP-PD

In this section, we present the Primal-Dual Representation Learning Algorithm (REP-PD) for low-rank CMDPs. REP-PD achieves provable safety guarantees and near-optimal policies by synergistically combining representation learning with a primal-dual policy optimization framework.

---

**Algorithm 1** Primal-Dual Representation Learning Algorithm (REP-PD)

---

1: **Input:** Parameters $\lambda_k, \alpha_k, \eta$, function class $\mathcal{M} = \{(\mu, \phi) : \mu \in \Theta, \phi \in \Gamma\}$, Iteration $K$
2: Initialize $\pi_0(\cdot \mid s)$ to be uniform, $Y_1 = 0$; set $\mathcal{D}_0 = \emptyset, \mathcal{D}'_0 = \emptyset$.
3: **for** episode $k = 1, \cdots, K$ **do**
4:     Collect the transition $(s, a, s', a', \tilde{s})$ where $s \sim d_{P^\star}^{\pi_{k-1}}, a \sim U(\mathcal{A}), s' \sim P^\star(\cdot|s, a), a' \sim U(\mathcal{A}), \tilde{s} \sim P^\star(\cdot|s', a')$, where $\mathcal{U}(\mathcal{A})$ denotes the uniform distribution on $\mathcal{A}$.
5:     Update datasets : $\mathcal{D}_k = \mathcal{D}_{k-1} + \{(s, a, s')\}, \quad \mathcal{D}'_k = \mathcal{D}'_{k-1} + \{(s', a', \tilde{s})\}$.
6:     Learn representation with MLE oracle:

$$\hat{P}_k := (\hat{\mu}_k, \hat{\phi}_k) = \arg \max_{(\mu, \phi) \in \mathcal{M}} \mathbb{E}_{\mathcal{D}_k + \mathcal{D}'_k} \left[ \ln \mu^\top(s') \phi(s, a) \right].$$

7:     Update empirical covariance matrix $\hat{\Sigma}_k = \sum_{s,a \in \mathcal{D}_k} \hat{\phi}_k(s, a) \hat{\phi}_k(s, a)^\top + \lambda_k I$.
8:     Set the exploration bonus: $\hat{b}_k(s, a) := \min \left( \alpha_k \sqrt{\hat{\phi}_k(s, a)^\top \hat{\Sigma}_k^{-1} \hat{\phi}_k(s, a)}, \bar{b} \right)$.
9:     Update policy $\pi_k = \arg \max_\pi Q_{\hat{P}_k, r+\hat{b}_k}^\pi + Y_k Q_{\hat{P}_k, g+\hat{b}_k}^\pi$.
10:    Update $Y_{k+1} = \max\{\min\{Y_k + \eta(b - V_{\hat{P}_k, g+\hat{b}_k}^{\pi_k}(s_1)), \xi\}, 0\}$.
11: **end for**
12: **Return** $\pi_1, \cdots, \pi_K$

---

### 3.1 REP-PD Algorithm

As illustrated in Algorithm 1, REP-PD operates within an iterative loop across $K$ episodes, with each iteration systematically refining the agent's environmental understanding and policy. This refinement is achieved through three tightly coupled phases: Environment Modeling via Representation Learning, Uncertainty-Aware Exploration, and Constraint-Guided Policy Optimization.

**Environment Modeling via Representation Learning:** The initial and foundational step in REP-PD is to overcome the challenge of unknown environment dynamics in high-dimensional state spaces. To this end, the algorithm first employs representation learning via MLE to approximate the underlying environment's transition function. By collecting transition data $(s, a, s', a', \tilde{s})$ via a data collection strategy that mixes the current policy with uniform exploration, REP-PD constructs empirical datasets. The datasets are then utilized to learn a low-rank transition model $\hat{P}_k = (\hat{\mu}_k, \hat{\phi}_k)$ by maximizing the log-likelihood of observed transitions. This process ensures the learned model effectively captures the inherent low-rank structure of the CMDP, providing an efficient and accurate approximation of the environment's dynamics without requiring prior feature knowledge.

**Uncertainty-Aware Exploration through Bonus Construction:** With an approximated transition representation in hand, the next critical challenge is to facilitate effective and safe exploration, particularly under the uncertainty of learned features. REP-PD addresses this by introducing an uncertainty-aware exploration bonus derived by the learned representations. Specifically, the algorithm computes an empirical covariance matrix, $\hat{\Sigma}_k$, from the collected state-action features, which quantifies the uncertainty associated with these features. An exploration bonus, $\hat{b}_k(s, a)$, is then constructed as a scaled Mahalanobis norm of $\hat{\phi}_k(s, a)$ with respect to $\hat{\Sigma}_k$, capped by an upper limit of $\bar{b}$ to ensure stability. This bonus will be integrated into the composite Q-function in the subsequent phase to encourage exploration. This is crucial for balancing the exploitation of current knowledge with the necessary exploration for robust representation learning and policy optimization.

**Constraint-Guided Policy Optimization with Primal-Dual Adaptation:** The final, crucial step integrates the learned environment model and the exploration strategy into a primal-dual policy optimization module to achieve the primary objective of maximizing rewards under safety constraints. The core idea is to update the policy $\pi_k$ by maximizing a composite Q-function: $\pi_k = \arg \max_\pi Q_{\hat{P}_k, r+\hat{b}_k}^\pi + Y_k Q_{\hat{P}_k, g+\hat{b}_k}^\pi$. This composite function merges the reward and utility Q-functions, each augmented with an exploration bonus derived from the learned features. This crucial augmentation seamlessly integrates insights from representation learning and exploration into the policy update mechanism.

Constraint satisfaction is managed by a Lagrangian multiplier $Y_k$, which adaptively weights the constraint within this composite objective. This Lagrangian multiplier is updated by taking a step toward minimizing $\mathcal{L}(\pi, Y)$ over $Y \geqslant 0$, following the rule: $Y_{k+1} = \max\{\min\{Y_k + \eta(b - V^{\pi_k}_{\hat{P}_k, g+\hat{b}_k}(s_1)), \xi\}, 0\}$, where $\eta > 0$ is the step-size and $\xi$ is an upper bound ensuring $Y^\star \in [0, \xi]$. If the estimated constraint violation $b - V^{\pi_k}_g(s_1)$ is non-negative, $Y_k$ increases to prioritize safety; otherwise, it decreases to focus on reward maximization. This adaptive Lagrangian multiplier adjustment ensures a dynamic trade-off between reward maximization and constraint adherence, driving the algorithm towards a policy that is both high-rewarding and compliant with safety requirements. The careful selection and interplay of these parameters are crucial for the method's theoretical guarantees, specifically ensuring near-optimal regret and bounded soft constraint violation that scales polynomially with the feature dimension rather than the state space dimension.

### 3.2 THEORETICAL ANALYSIS

In this subsection, we conduct an analysis of REP-PD, establishing theoretical bounds on its regret and soft constraint violation. Furthermore, we demonstrate that it is possible to achieve zero soft constraint violation for large enough $K$.

**Theorem 6** (Regret and Soft Violation Bounds). *Set* $\alpha_k = \mathcal{O}(\sqrt{(|\mathcal{A}| + d^2)\gamma \ln(|\mathcal{M}|k/\delta)})$, $\lambda_k = \mathcal{O}(d\ln(|\mathcal{M}|k/\delta))$, $\eta = \frac{2(1-\gamma)}{\theta \bar{b} \sqrt{K}}$ *in algorithm 1. With probability* $1 - \delta$, *we have*

$$Regret(K) \leqslant \tilde{\mathcal{O}}\left(\frac{|\mathcal{A}|\bar{b}d^2\sqrt{K\ln(K|\mathcal{M}|/\delta)}}{\theta(1-\gamma)^2}\right), Violation_{soft}(K) \leqslant \tilde{\mathcal{O}}\left(\frac{|\mathcal{A}|\bar{b}d^2\sqrt{K\ln(K|\mathcal{M}|/\delta)}}{\theta(1-\gamma)^2}\right).$$

Thus, we obtain a bound that is independent of the state space dimension, while exhibits polynomial dependency on $\theta, |\mathcal{A}|, \bar{b}, d, \ln(|\mathcal{M}|/\delta)$ and $1/(1-\gamma)$.

**Remark 7** (Tightness of Bounds). *The* $\tilde{\mathcal{O}}(|\mathcal{A}|d^2\sqrt{K})$ *scaling matches the lower bound for reward known unconstrained low-rank MDPs (21), suggesting our constraint-aware extensions avoid excessive overhead. The* $\theta^{-1}$ *dependency aligns with Slater's condition and is unavoidable in constrained optimization (19).*

The regret and soft constraint violation bounds presented in Theorem 6 are supported by three fundamental components, with the complete proof detailed in the appendix:

1. **Representation Learning Guarantees:** This component (lemma 14-17) ensures that the learned features $\hat{\phi}_k$ accurately approximate the true low-rank structure $\phi^\star$, with a statistical error propagating polynomially in the feature dimension $d$ (not state dimension $|\mathcal{S}|$).

2. **Lagrangian Stability under Slater's Condition:** Through primal-dual policy optimization (Lemma 3,12-13), the composite value function $V^{\pi, Y}$ combines reward and utility objectives, enabling a unified optimization landscape. The dual variable $Y_k$ adaptively penalizes constraint violations, ensuring $Y_k$ remains bounded and balances exploration and constraint adherence.

3. **Elliptical Potential-based Regret and Violation Aggregation:** Using the simulation lemma (Lemma 30), we decompose the composite regret into terms involving the bonus $\hat{b}_k$ and the representaion error. The Mahalanobis norm $\|\hat{\phi}_k\|_{\hat{\Sigma}_k^{-1}}$ quantifies uncertainty, and its sum over episodes is bounded via elliptical potential lemma (Lemma 29).

**Remark 8** (Comparison to Linear CMDPs). *Our work eliminates the requirement of known* $\phi^\star$ *in linear CMDPs. The cost is an additional* $|\mathcal{A}|$ *factor in regret and violation bounds, reflecting the overhead of learning* $\hat{\phi}$ *online, which is a reasonable trade-off. Furthermore, our theoretical analysis diverges from prior linear CMDP approaches (15; 18) by circumventing the need for pointwise transition error bounds. Instead, we exploit the low-rank structure to relate global representation error to policy suboptimality.*

Moreover, we can achieve zero soft constraint violation for sufficiently large $K$.

---

**Algorithm 2** Primal-Dual Representation Learning Algorithm with hard constraint violation (REP-PD-hard)

---

1: **Input:** Parameters $\lambda_k, \alpha_k, \tau, \eta$, Model class $\mathcal{M} = \{(\mu, \phi) : \mu \in \Theta, \phi \in \Gamma\}$, Iteration $K$.
2: Initialize $\pi_0(\cdot \mid s)$ to be uniform; set $\mathcal{D}_0 = \emptyset, \mathcal{D}'_0 = \emptyset$
3: **for** episode $k = 1, \cdots, K$ **do**
4:    Collect the transition $(s, a, s', a', \tilde{s})$ where $s \sim d_{P^\star}^{\pi_{k-1}}, a \sim U(\mathcal{A}), s' \sim P^\star(\cdot|s,a), a' \sim U(\mathcal{A}), \tilde{s} \sim P^\star(\cdot|s',a')$, where $\mathcal{U}(\mathcal{A})$ denotes the uniform distribution on $\mathcal{A}$.
5:    Update datasets : $\mathcal{D}_k = \mathcal{D}_{k-1} + \{(s, a, s')\}, \quad \mathcal{D}'_k = \mathcal{D}'_{k-1} + \{(s', a', \tilde{s})\}$.
6:    Learn representation with MLE oracle:

$$\hat{P}_k := (\hat{\mu}_k, \hat{\phi}_k) = \arg \max_{(\mu, \phi) \in \mathcal{M}} \mathbb{E}_{\mathcal{D}_k + \mathcal{D}'_k} \left[ \ln \mu^\top(s') \phi(s, a) \right].$$

7:    Update empirical covariance matrix $\hat{\Sigma}_k = \sum_{(s,a) \in \mathcal{D}_k} \hat{\phi}_k(s, a) \hat{\phi}_k(s, a)^\top + \lambda_k I$.
8:    Set the exploration bonus: $\hat{b}_k(s, a) := \min \left( \alpha_k \sqrt{\hat{\phi}_k(s, a)^\top \hat{\Sigma}_k^{-1} \hat{\phi}_k(s, a)}, \bar{b} \right)$
9:    Initialize $Y_k = 0$.
10:   **while** $Y_k \leqslant K^{1/4}$ **do**
11:       Set $\hat{\pi}_k = \arg \max_\pi Q_{\hat{P}_k, r+\hat{b}_k}^\pi + Y_k Q_{\hat{P}_k, g+\hat{b}_k}^\pi$.
12:       Update $\pi_k = \text{soft-max}_\tau (Q_{\hat{P}_k, r+\hat{b}_k}^{\hat{\pi}_k} + Y_k Q_{\hat{P}_k, g+\hat{b}_k}^{\hat{\pi}_k})$ and $Y_k = Y_k + \eta$.
13:       **if** $V_g^{\pi_k} \geqslant b$:    **break**.
14:   **end while**
15:   **if** $Y_k > K^{1/4}$:    Set $Y_k = K^{1/4}$.
16: **end for**
17: **Return** $\pi_1, \cdots, \pi_K$

---

**Corollary 9.** *In Algorithm 1, replacing $b = b + \zeta$, and set $\xi = \frac{4}{(1-\gamma)\theta}$. Then, with probability at least $1 - p$, we have:*

$$Regret(K) \leqslant \mathcal{O} \left( \frac{|\mathcal{A}|\bar{b}d^2\sqrt{K}}{\theta^2(1-\gamma)^3} \right), \quad Violation_{soft}(K) \leqslant \max \left\{ \mathcal{O} \left( \frac{|\mathcal{A}|\bar{b}d^2\sqrt{K}}{\theta(1-\gamma)^2} \right) - K\zeta, 0 \right\},$$

*where $\zeta = \min\{\mathcal{O} \left( \frac{|\mathcal{A}|\bar{b}d^2\sqrt{K}}{K\theta(1-\gamma)^2} \right), \theta/2\}$.*

We refer to this variant of the algorithm as REP-PD-0. When $\mathcal{O} \left( \frac{|\mathcal{A}|d^2\sqrt{K}}{K\theta(1-\gamma)^2} \right) \leqslant \theta/2$, the violation term becomes non-positive, thereby ensuring zero soft constraint violation for sufficiently large $K$. However, the upper bound on regret increases to $\mathcal{O} \left( \frac{|\mathcal{A}|d^2\sqrt{K}}{\theta^2(1-\gamma)^3} \right)$ as a consequence. This trade-off aligns with findings in prior work on tabular CMDPs (49) and linear CMDPs (15), which demonstrate that achieving zero violation inherently incurs an additional $1/(1 - \gamma)$ factor. Despite our distinct setting, this additional $1/(1 - \gamma)$ dependence appears to be similarly unavoidable. A detailed analysis of zero soft constraint violation is provided in the appendix.

## 4 REP-PD-HARD

In this section, we delve into the more stringent scenario of hard constraint violation. We introduce the Primal-Dual Representation Learning Algorithm for hard constraint violation, referred to as REP-PD-hard, that achieves sublinear regret and hard constraint violation guarantees by designing a novel policy optimization module.

### 4.1 REP-PD-HARD ALGORITHM

As discussed in the Background, hard constraint violation is a stricter metric than soft constraint violation. Consequently, designing algorithms that simultaneously achieve sub-linear regret and maintain bounded hard constraint violation with unknown dynamics becomes considerably more challenging. To address this problem, REP-PD-hard incorporates a crucially distinct policy optimization module. The core innovation is to find the perfect dual variable at each episode to balance reward maximization and constraint satisfaction.

As shown in Algorithm 2, REP-PD-hard maintains an iterative loop, alternating between representation learning, constraint-aware exploration, and primal-dual policy optimization. The key difference between REP-PD-hard and REP-PD lies in the primal-dual policy optimization module. To satisfy the stricter hard constraint violation, REP-PD-hard must find the perfect dual variable in each episode. Moreover, instead of the greedy policy used in REP-PD, REP-PD-hard employs a soft-max policy. This choice is motivated by the distinct analysis required for hard constraint violation, which leverages the Lipschitz property of the soft-max policy. The softmax policy soft-max$(X) = \{\text{soft-max}_\tau^a(X)\}_{a=1}^{|\mathcal{A}|}$ for any $X \in \mathbb{R}^{|\mathcal{A}|}$ is a $|\mathcal{A}|$-dimensional vector with parameter $\tau$ where the $a$-th component

$$\text{soft-max}_\tau^a(X) = \frac{\exp(X_a/\tau)}{\sum_{i=1}^{|\mathcal{A}|} \exp(X_i/\tau)}.$$

The algorithm iteratively update the policy and increase $Y_k$ by $\eta$ until $Y_k \geqslant K^{1/4}$ or $V_g^{\pi_k} \geqslant b$. Upon exiting the inner loop, one of the following two cases holds:

- $Y_k = K^{1/4}$ and $V_g^{\pi_k}(s_1) < b$. In this case, we reach the upper bound of the dual variable. We demonstrate that the upper bound of $K^{1/4}$ is sufficient to achieve $\tilde{O}(K^{3/4})$ regret and $\tilde{O}(\sqrt{K})$ hard constraint violation.

- $Y_k \leqslant K^{1/4}$ and $V_g^{\pi_k} \geqslant b$. In this case, we prove that with an appropriate choice of $\eta$ and $\tau$, we can ensure $V_g^{\pi_k}(s_1) \leqslant b + \mathcal{O}(K^{-1})$, thus balancing regret and hard constraint violation.

**Remark 10** (Relaxed Assumptions in REP-PD-Hard). *Compared to existing CMDPs algorithms with bounded hard constraint violation, REP-PD-Hard significantly relaxes standard assumptions. Beyond unknown dynamics, it removes the requirement of a known safe action set, which is commonly assumed in prior work (13; 14; 18).*

## 4.2 THEORETICAL ANALYSIS

Due to the stricter requirement of hard constraint violation, the analysis of REP-PD-hard differs with that of REP-PD. Simply put, we categorize the episodes into two cases based on the value of $Y_k$: episodes where $Y_k < \sqrt{K}$ and episodes where $Y_k = \sqrt{K}$. As previously discussed, by carefully selecting the values of $\eta$ and $\tau$, we ensure that both the regret and hard constraint violation are bounded for both cases. The complete proof is provided in Appendix.

We prove Algorithm 2 achieves the regret and hard constraint violation which are sublinear in $K$.

**Theorem 11** (Regret and Hard Violation Bounds). *Set $\alpha_k = O(\sqrt{(|\mathcal{A}| + d^2)\gamma \ln(|\mathcal{M}|k/\delta)})$, $\lambda_k = O\left(d \ln(|\mathcal{M}|k/\delta)\right)$, $\tau = \frac{\sqrt{K}}{(1-\gamma)^3}$, $\eta = \frac{1}{\sqrt{K}(1-\gamma)}$. in algorithm 2 With probability $1 - \delta$, we have*

$$Regret(K) \leqslant \mathcal{O}\left(\frac{|\mathcal{A}|\bar{b}d^2 K^{3/4}\ln(|\mathcal{M}|/\delta)}{(1-\gamma)^2}\right), Violation_{hard}(K) \leqslant \mathcal{O}\left(\frac{|\mathcal{A}|\bar{b}d^2\sqrt{K}\ln(|\mathcal{M}|/\delta)}{(1-\gamma)^2}\right).$$

To our knowledge, this is the first result that achieves sublinear regret and hard constraint violation bound for low-rank CMDPs. The $\tilde{O}(\sqrt{K})$ hard constraint violation bound matches the bounds attained in the tabular and linear setting. The $K^{3/4}$ regret suboptimality stems from the exploration due to unknown transition dynamics. These results confirm that REP-PD-Hard maintains strong theoretical guarantees under significantly weaker assumptions, marking a novel advancement in constrained RL with unknown dynamics.

## 5 CONCLUSION

We propose the first provably efficient algorithm for representation learning in CMDPs without requiring known latent features. Our approach interleaves representation learning, exploration bonuses, and primal-dual optimization to ensure both near-optimal reward and bounded soft constraint violation. We further propose REP-PD-hard to handle stricter hard constraints while maintaining sublinear regret. Together, our results establish a scalable and theoretically grounded framework for constrained reinforcement learning in high-dimensional and unknown environments. A key direction is extending our results to other safety type, such as instantaneous safety. Utilizing a decaying $\epsilon$-greedy strategy for exploration is also a direction for future work.

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

## A  NOTATIONS

Firstly, we summarize the notations frequently used in the proofs. Hereafter, we assume $c_0, c_1, \cdots,$ are some universal constants, and the notation $f(x) \lesssim g(x)$ means there exists some constant $c_1 > 0$, such that $f(x) \leqslant c_1 g(x)$ for any $x$.

Let $d_{P*}^{\pi}(s)$ be the state occupancy measure

$$d_{P*}^{\pi}(s) = (1-\gamma)\sum_{h=1}^{\infty}\gamma^h P_{\pi}^{\star}(s_h = s | s_1 \sim d_1),$$

where $P_{\pi}^*(s_h = s | s_1 \sim d_1)$ is the probability that $s_h = s$, after starting at state $s_1 \sim d_1$ and following $\pi$ thereafter.

Also, let $d_{P*}^{\pi}(s,a) = \pi(a|s)d_{P*}^{\pi}(s)$ be the state-action occupation measure. Hence,

$$V_j^{\pi}(s_1) = \int_{s,a} j(s,a) dd_{P*}^{\pi}(s,a). \tag{7}$$

We define

$$\rho_k(s) := \frac{1}{k}\sum_{i=0}^{k-1} d_{P^{\star}}^{\pi_i}(s).$$

With slight abuse of notation, we overload the above notation and use $\rho_k$ for $1/k \sum_{i=0}^{k-1} d_{P^{\star}}^{\pi_i}(s,a)$. Next, define $\rho_k' : \mathcal{S} \to \mathbb{R}$ as a marginal distribution of $s'$ for a triple

$$(s,a,s') \sim \rho_k(s)U(a)P^{\star}(s' \mid s,a).$$

We define three matrices as follows:

$$\Sigma_{\rho_k \times U(\mathcal{A}),\phi} = k\mathbb{E}_{s \sim \rho_k, a \sim U(\mathcal{A})}[\phi(s,a)\phi^{\top}(s,a)] + \lambda_k I,$$
$$\Sigma_{\rho_k,\phi} = k\mathbb{E}_{(s,a)\sim\rho_k}[\phi(s,a)\phi^{\top}(s,a)] + \lambda_k I,$$
$$\hat{\Sigma}_{k,\phi} = k\mathbb{E}_{(s,a)\sim\mathcal{D}_k}[\phi(s,a)\phi^{\top}(s,a)] + \lambda_k I.$$

Note that for a fixed $\phi$, $\hat{\Sigma}_{k,\phi}$ is an unbiased estimate of $\Sigma_{\rho_k \times U(\mathcal{A}),\phi}$.

## B  PROOF OF THEOREM 6

We now provide a detailed proof of Theorem 6, which establishes the regret and soft constraint violation bounds for REP-PD. We start by proving a lemma that bounds the dual variable updates:

**Lemma 12.** *For $Y \in [0,\xi]$, we have*

$$\sum_{k=1}^{K}(Y - Y_k)(b - V_{\hat{P}_k, g+\hat{b}_k}^{\pi_k}(s_1)) \leqslant \frac{Y^2}{2\eta} + \frac{\eta\bar{b}^2 K}{2(1-\gamma)^2} \tag{8}$$

*Proof.*

$$|Y_{k+1} - Y|^2$$
$$= |Proj_{[0,\xi]}(Y_k + \eta(b - V_{\hat{P}_k, g+\hat{b}_k}^{\pi_k}(s_1))) - Proj_{[0,\xi]}(Y)|^2$$
$$\leqslant (Y_k + \eta(b - V_{\hat{P}_k, g+\hat{b}_k}^{\pi_k}(s_1)) - Y)^2$$
$$\leqslant (Y_k - Y)^2 + \frac{\eta^2\bar{b}^2}{(1-\gamma)^2} + 2\eta(Y_k - Y)(b - V_{\hat{P}_k, g+\hat{b}_k}^{\pi_k}(s_1))$$

Summing over $k$, we obtain

$$0 \leqslant |Y_{k+1} - Y|^2$$
$$\leqslant |Y_1 - Y|^2 + 2\eta\sum_{k=1}^{K}(b - V_{\hat{P}_k, g+\hat{b}_k}^{\pi_k}(s_1))(Y_k - Y) + \eta^2\bar{b}^2 K/(1-\gamma)^2$$

Thus, we have

$$\sum_{k=1}^{K}(Y-Y_k)(b-V^{\pi_k}_{\hat{P}_k,g+\hat{b}_k}(s_1)) \leqslant \frac{|Y_1-Y|^2}{2\eta} + \frac{\eta\bar{b}^2 K}{2(1-\gamma)^2}.$$

Since $Y_1 = 0$, we have the result. $\qquad\square$

Using Lemma 12, we can prove the following lemma:

**Lemma 13.** *For $Y \in [0, \xi]$, we have*

$$\sum_{k=1}^{K}(V^{\pi^*}_{P^\star,r}(s_1)-V^{\pi_k}_{P^\star,r}(s_1))+Y\sum_{k=1}^{K}(b-V^{\pi_k}_{P^\star,g}(s_1))$$

$$\leqslant \sum_{k=1}^{K}(V^{\pi^*}_{P^\star,r}(s_1)-V^{\pi_k}_{P^\star,r}(s_1)) + \frac{Y^2}{2\eta} + \frac{\eta^2\bar{b}^2 K}{2(1-\gamma)^2}$$

$$+\sum_{k=1}^{K}Y_k(V^{\pi^*}_{P^\star,g}(s_1)-V^{\pi_k}_{P^\star,g}(s_1)) + \sum_{k=1}^{K}(Y-Y_k)(V^{\pi_k}_{\hat{P}_k,g+\hat{b}_k}(s_1)-V^{\pi_k}_{P^\star,g}(s_1))$$

*Proof.*

$$Y\sum_{k=1}^{K}(b-V^{\pi_k}_{P^\star,g}(s_1))$$

$$=\sum_{k=1}^{K}(Y-Y_k)(b-V^{\pi_k}_{\hat{P}_k,g+\hat{b}_k}(s_1))+\sum_{k=1}^{K}Y_k(b-V^{\pi_k}_{\hat{P}_k,g+\hat{b}_k}(s_1))$$

$$+\sum_{k=1}^{K}(Y-Y_k)(V^{\pi_k}_{\hat{P}_k,g+\hat{b}_k}(s_1)-V^{\pi_k}_{P^\star,g}(s_1))$$

$$\leqslant \frac{Y^2}{2\eta} + \frac{\eta^2\bar{b}^2 K}{2(1-\gamma)^2} + \sum_{k=1}^{K}Y_k(b-V^{\pi_k}_g(s_1))$$

$$+\sum_{k=1}^{K}(Y-Y_k)(V^{\pi_k}_{\hat{P}_k,g+\hat{b}_k}(s_1)-V^{\pi_k}_{P^\star,g}(s_1))$$

$$\leqslant \frac{Y^2}{2\eta} + \frac{\eta^2\bar{b}^2 K}{2(1-\gamma)^2} + \sum_{k=1}^{K}Y_k(V^{\pi^*}_{P^\star,g}(s_1)-V^{\pi_k}_g(s_1))$$

$$+\sum_{k=1}^{K}(Y-Y_k)(V^{\pi_k}_{\hat{P}_k,g+\hat{b}_k}(s_1)-V^{\pi_k}_{P^\star,g}(s_1))$$

where first inequality follows from lemma 12 and the second inequality follows from the fact that $V^{\pi^*}_g(s_1) \geqslant b$.

Hence, we have that

$$\sum_{k=1}^{K}(V^{\pi^*}_{P^\star,r}(s_1)-V^{\pi_k}_{P^\star,r}(s_1)) + Y\sum_{k=1}^{K}(b-V^{\pi_k}_{P^\star,g}(s_1))$$

$$\leqslant \frac{Y^2}{2\eta} + \frac{\eta^2\bar{b}^2 K}{2(1-\gamma)^2} + \sum_{k=1}^{K}(V^{\pi^*}_{P^\star,r}(s_1)-V^{\pi_k}_{P^\star,r}(s_1))$$

$$+\sum_{k=1}^{K}Y_k(V^{\pi^*}_{P^\star,g}(s_1)-V^{\pi_k}_{P^\star,g}(s_1)) + \sum_{k=1}^{K}(Y-Y_k)(V^{\pi_k}_{\hat{P}_k,g+\hat{b}_k}(s_1)-V^{\pi_k}_{P^\star,g}(s_1))$$

$$\square$$

Then we provide an important lemma to ensure the concentration of the bonus term.

**Lemma 14** (Concentration of the bonus term). *Set* $\lambda_k = \Theta(d \ln(k|\Phi|/\delta))$ *for any* $k$. *Define* $\Sigma_{\rho_k,\phi} = k\mathbb{E}_{s\sim\rho_k,a\sim U(\mathcal{A})}[\phi(s,a)\phi^\top(s,a)] + \lambda_k I$, $\quad \hat{\Sigma}_{k,\phi} = \sum_{i=0}^{k-1} \phi(s^{(i)},a^{(i)})\phi^\top(s^{(i)},a^{(i)}) + \lambda_k I$. *With probability* $1-\delta$, *we have* $\forall k \in \mathbb{K}^+, \forall \phi \in \Phi$,

$$c_1\|\phi(s,a)\|_{\Sigma_{\rho_k \times U(\mathcal{A}),\phi}^{-1}} \leqslant \|\phi(s,a)\|_{\hat{\Sigma}_{k,\phi}^{-1}} \leqslant c_2\|\phi(s,a)\|_{\Sigma_{\rho_k \times U(\mathcal{A}),\phi}^{-1}}.$$

*Proof.* See ((21), Lemma 11) $\qquad\qquad\qquad\qquad\qquad\qquad\qquad\qquad\qquad\qquad\qquad\quad\square$

**Lemma 15** (One-step back inequality for the learned model). *Take any* $g \in \mathcal{S} \times \mathcal{A} \to \mathbb{R}$ *such that* $\|g\|_\infty \leqslant B$. *We condition on the event where the MLE guarantee (27):*

$$\mathbb{E}_{s\sim\rho_k,a\sim U(\mathcal{A})}[f_k(s,a)] \lesssim \zeta_k,$$

*holds. Then, for any policy* $\pi$, *we have*

$$|\mathbb{E}_{(s,a)\sim d_{\hat{P}_k}^\pi}[g(s,a)]| \leqslant \mathbb{E}_{(\tilde{s},\tilde{a})\sim d_{\hat{P}_k}^\pi}\|\hat{\phi}_k(\tilde{s},\tilde{a})\|_{\Sigma_{\rho_k \times U(\mathcal{A}),\hat{\phi}_k}^{-1}} \cdot$$
$$\sqrt{\left\{k|\mathcal{A}|\mathbb{E}_{s\sim\rho_k',a\sim U(\mathcal{A})}[g^2(s,a)]\right\} + B^2\lambda_k d + kB^2\zeta_k} + \sqrt{(1-\gamma)|\mathcal{A}|\mathbb{E}_{s\sim\rho_k,a\sim U(\mathcal{A})}[g^2(s,a)]}.$$

Recall $\Sigma_{\rho_k \times U(\mathcal{A}),\hat{\phi}_k} = k\mathbb{E}_{s\sim\rho_k,a\sim U(\mathcal{A})}[\hat{\phi}_k(s,a)\hat{\phi}_k^\top(s,a)] + \lambda_k I$.

*Proof.* First, we have an equality:

$$\mathbb{E}_{(s,a)\sim d_{\hat{P}_k}^\pi}[g(s,a)] = \gamma\mathbb{E}_{(\tilde{s},\tilde{a})\sim d_{\hat{P}_k}^\pi,s\sim\hat{P}_k(\tilde{s},\tilde{a}),a\sim\pi(s)}[g(s,a)] + (1-\gamma)\mathbb{E}_{s\sim d_1,a\sim\pi(s_1)}[g(s,a)], \tag{9}$$

The second term in 9 is upper-bounded by

$$(1-\gamma)\sqrt{\max_{(s,a)} \frac{d_1(s)\pi(a\mid s)}{\rho_k(s)u(a)}\mathbb{E}_{s\sim\rho_k,a\sim U(\mathcal{A})}[g^2(s,a)]}$$
$$\leqslant (1-\gamma)\sqrt{\max_{(s,a)} \frac{d_1(s)\pi(a\mid s)}{(1-\gamma)d_1(s)u(a)}\mathbb{E}_{s\sim\rho_k,a\sim U(\mathcal{A})}[g^2(s,a)]}$$
$$\leqslant \sqrt{(1-\gamma)|\mathcal{A}|\mathbb{E}_{s\sim\rho_k,a\sim U(\mathcal{A})}[g^2(s,a)]}.$$

Next we consider the first term in (9). By CS inequality, we have

$$\mathbb{E}_{(\tilde{s},\tilde{a})\sim d_{\hat{P}_k}^\pi,s\sim\hat{P}_k(\tilde{s},\tilde{a}),a\sim\pi(s)}[g(s,a)]$$
$$= \mathbb{E}_{(\tilde{s},\tilde{a})\sim d_{\hat{P}_k}^\pi}\hat{\phi}_k(\tilde{s},\tilde{a})^\top \int \sum_a \hat{\mu}_k(s)\pi(a\mid s)g(s,a)ds$$
$$\leqslant \mathbb{E}_{(\tilde{s},\tilde{a})\sim d_{\hat{P}_k}^\pi}\|\hat{\phi}_k(\tilde{s},\tilde{a})\|_{\Sigma_{\rho_k \times U(\mathcal{A}),\hat{\phi}_k}^{-1}} \cdot \left\|\int \sum_a \hat{\mu}_k(s)\pi(a\mid s)g(s,a)d(s)\right\|_{\Sigma_{\rho_k \times U(\mathcal{A}),\hat{\phi}_k}}.$$

Then,

$$\| \int \sum_a \hat\mu_k(s)\pi(a \mid s)g(s,a)d(s)\|^2_{\Sigma_{\rho_k \times U(\mathcal{A}),\hat\phi_k}}$$

$$\leqslant \left\{ \int \sum_a \hat\mu_k(s)\pi(a \mid s)g(s,a)d(s)\right\}^\top$$

$$\left\{ k\mathbb{E}_{s\sim\rho_k,a\sim U(\mathcal{A})}[\hat\phi_k\hat\phi_k^\top] + \lambda_k I\right\} \left\{ \int \sum_a \hat\mu_k(s)\pi(a \mid s)g(s,a)d(s)\right\}$$

$$\leqslant k\mathbb{E}_{\tilde s\sim\rho_k,\tilde a\sim U(\mathcal{A})}\left\{ \left[\int \sum_a \hat\mu_k(s)^\top\hat\phi_k(\tilde s,\tilde a)\pi(a\mid s)g(s,a)d(s)\right]^2\right\} + B^2\lambda_k d$$

$$= k\mathbb{E}_{\tilde s\sim\rho_k,\tilde a\sim U(\mathcal{A})}\left[\left(\mathbb{E}_{s\sim\hat P_k(\tilde s,\tilde a),a\sim\pi(s)}[g(s,a)]\right)^2\right] + B^2\lambda_k d$$

$$\leqslant k\mathbb{E}_{s\sim\rho_k,a\sim U(\mathcal{A})}\left[\left(\mathbb{E}_{s\sim P^\star(\tilde s,\tilde a),a\sim\pi(s)}[g(s,a)]\right)^2\right] + B^2\lambda_k d + kB^2\zeta_k \qquad \text{(MLE guarantee)}$$

$$\leqslant k\mathbb{E}_{\tilde s\sim\rho_k,\tilde a\sim U(\mathcal{A}),s\sim P^\star(\tilde s,\tilde a),a\sim\pi(s)}\left[g^2(s,a)\right] + B^2\lambda_k d + B^2 k\zeta_k. \qquad \text{(Jensen)}$$

$$\leqslant k|\mathcal{A}| \left\{\mathbb{E}_{\tilde s\sim\rho_k,\tilde a\sim U(\mathcal{A}),s\sim P^\star(\tilde s,\tilde a),a\sim U(\mathcal{A})}\left[g^2(s,a)\right]\right\} + B^2\lambda_k d + B^2 k\zeta_k$$

$$\text{(Importance sampling)}$$

$$\leqslant k|\mathcal{A}|\mathbb{E}_{s\sim\rho'_k,a\sim U(\mathcal{A})}\left[g^2(s,a)\right] + B^2\lambda_k d + B^2 k\zeta_k. \qquad \text{(Definition of } \rho'_k)$$

Here we use $\| \sum_a \pi(a \mid s)g(s,a)\|_\infty \leqslant B$ and $\int \|\hat\mu_k(s)h(s)d(s)\|_2 \leqslant \sqrt{d}$ for any $h : \mathcal{S} \to [0,1]$ in the second inequality.

Then, the final statement is immediately concluded. $\qquad\square$

Now, we provide one-step back inequality for the true model and learned model.

**Lemma 16** (One-step back inequality for the true model ). *Take any $g \in \mathcal{S} \times \mathcal{A} \to \mathbb{R}$ such that $\|g\|_\infty \leqslant B$. Then,*

$$\mathbb{E}_{(s,a)\sim d^\pi_{P^\star}}[g(s,a)] \leqslant \mathbb{E}_{(\tilde s,\tilde a)\sim d^\pi_{P^\star}}\|\phi^\star(\tilde s,\tilde a)\|_{\Sigma^{-1}_{\rho_k,\phi^\star}}\sqrt{\gamma}\cdot\sqrt{k|\mathcal{A}|\mathbb{E}_{s\sim\rho_k,a\sim U(\mathcal{A})}[g^2(s,a)] + \lambda_k dB^2}$$

$$+ \sqrt{(1-\gamma)|\mathcal{A}|\mathbb{E}_{s\sim\rho_k,a\sim U(\mathcal{A})}[g^2(s,a)]}.$$

Recall $\Sigma_{\rho_k,\phi^\star} = k\mathbb{E}_{(s,a)\sim\rho_k}[\phi^\star(s,a)\phi^\star(s,a)^\top] + \lambda_k I$.

*Proof.* First, we have

$$\mathbb{E}_{(s,a)\sim d^\pi_{P^\star}}[g(s,a)] = \gamma\mathbb{E}_{(\tilde s,\tilde a)\sim d^\pi_{P^\star},s\sim P^\star(\tilde s,\tilde a),a\sim\pi(s)}[g(s,a)] + (1-\gamma)\mathbb{E}_{s\sim d_1,a\sim\pi(s_0)}[g(s,a)]. \tag{10}$$

The second term in 10 is upper-bounded by

$$(1-\gamma)\sqrt{\max_{(s,a)}\frac{d_1(s)\pi(a\mid s)}{\rho_k(s)u(a)}\mathbb{E}_{s\sim\rho_k,a\sim U(\mathcal{A})}[g^2(s,a)]} \leqslant \sqrt{|\mathcal{A}|\mathbb{E}_{s\sim\rho_k,a\sim U(\mathcal{A})}[g^2(s,a)](1-\gamma)}.$$

By CS inequality, the first term in 10 is further bounded as follows:

$$\mathbb{E}_{(\tilde s,\tilde a)\sim d^\pi_{P^\star},s\sim P^\star(\tilde s,\tilde a),a\sim\pi(s)}[g(s,a)]$$

$$= \mathbb{E}_{(\tilde s,\tilde a)\sim d^\pi_{P^\star}}\phi^\star(\tilde s,\tilde a)^\top\int\sum_a \mu^\star(s)\pi(a\mid s)g(s,a)d(s)$$

$$\leqslant \mathbb{E}_{(\tilde s,\tilde a)\sim d^\pi_{P^\star}}\|\phi^\star(\tilde s,\tilde a)\|_{\Sigma^{-1}_{\rho_k,\phi^\star}}\left\|\int\sum_a \mu^\star(s)\pi(a\mid s)g(s,a)d(s)\right\|_{\Sigma_{\rho_k,\phi^\star}}.$$

Here, we have

$$\| \int \sum_a \mu^\star(s)\pi(a \mid s)g(s,a)d(s)\|^2_{\Sigma_{\rho_k,\phi^\star}}$$

$$\leqslant \left\{ \int \sum_a \mu^\star(s)\pi(a \mid s)g(s,a)d(s) \right\}^\top$$

$$\left\{ k\mathbb{E}_{(s,a)\sim\rho_k}[\phi^\star(s,a)\{\phi^\star(s,a)\}^\top] + \lambda_k I \right\} \left\{ \int \sum_a \mu^\star(s)\pi(a \mid s)g(s,a)d(s) \right\}$$

$$\leqslant k\mathbb{E}_{(\tilde{s},\tilde{a})\sim\rho_k}\left\{ \left[ \int \sum_a \mu^\star(s)^\top \phi^\star(\tilde{s},\tilde{a})\pi(a\,|\,s)g(s,a)d(s) \right]^2 \right\} + \lambda_k dB^2$$

$$\leqslant k\left\{ \mathbb{E}_{(\tilde{s},\tilde{a})\sim\rho_k,s\sim P^\star(\tilde{s},\tilde{a}),a\sim\pi(s)}\left[ g^2(s,a) \right] \right\} + \lambda_k dB^2. \qquad \text{(Jensen)}$$

Therefore,

$$k\left\{ \mathbb{E}_{(\tilde{s},\tilde{a})\sim\rho_k,s\sim P^\star(\tilde{s},\tilde{a}),a\sim\pi(s)}\left[ g^2(s,a) \right] \right\} + \lambda_k dB$$

$$\leqslant k|\mathcal{A}|\left\{ \mathbb{E}_{(\tilde{s},\tilde{a})\sim\rho_k,s\sim P^\star(\tilde{s},\tilde{a}),a\sim U(\mathcal{A})}\left[ g^2(s,a) \right] \right\} + \lambda_k dB^2 \qquad \text{(Importance sampling)}$$

$$\leqslant k|\mathcal{A}|\left\{ \frac{1}{\gamma}\mathbb{E}_{s\sim\rho_k,a\sim U(\mathcal{A})}\left[ g^2(s,a) \right] \right\} + \lambda_k dB^2.$$

In the last line, we use the following inequality:

$$\mathbb{E}_{s\sim\rho_k,a\sim U(\mathcal{A})}\left[ g^2(s,a) \right]$$

$$= \gamma\mathbb{E}_{(\tilde{s},\tilde{a})\sim\rho_k,s\sim P^\star(\tilde{s},\tilde{a}),a\sim U(\mathcal{A})}\left[ g^2(s,a) \right] + (1-\gamma)\mathbb{E}_{s_1\sim d_1,a\sim U(\mathcal{A})}\left[ g^2(s,a) \right]$$

$$\geqslant \gamma\mathbb{E}_{(\tilde{s},\tilde{a})\sim\rho_k,s\sim P^\star(\tilde{s},\tilde{a}),a\sim U(\mathcal{A})}\left[ g^2(s,a) \right].$$

Finally, we have

$$\mathbb{E}_{(s,a)\sim d^\pi_{P^\star}}\left[ g(s,a) \right]$$

$$\leqslant \mathbb{E}_{(\tilde{s},\tilde{a})\sim d^\pi_{P^\star}}\|\phi^\star(\tilde{s},\tilde{a})\|_{\Sigma^{-1}_{\rho_k,\phi^\star}}\sqrt{\gamma\left\{ k|\mathcal{A}|\mathbb{E}_{s\sim\rho_k,a\sim U(\mathcal{A})}\left[ g^2(s,a) \right] + \lambda_k dB^2 \right\}}$$

$$+ \sqrt{|\mathcal{A}|\mathbb{E}_{s\sim\rho_k,a\sim U(\mathcal{A})}\left[ g^2(s,a) \right](1-\gamma)}.$$

This concludes the proof. $\qquad\square$

Next, we prove the optimism in the initial distribution.

**Lemma 17** (Almost Optimism at the Initial Distribution). *Consider an episode $k$ $(1 \leqslant k \leqslant K)$ and set $\alpha_k = O(\sqrt{(|\mathcal{A}| + d^2)\gamma\ln(|\mathcal{M}|k/\delta)})$, $\lambda_k = O(d\ln(|\mathcal{M}|k/\delta))$, $\zeta_k = O\left( \frac{\ln(|\mathcal{M}|k/\delta)}{k} \right)$. With probability $1 - \delta$, $\forall k \in [1, \cdots, K], \forall \pi \in \Pi$, we have*

$$V^\pi_{\hat{P}_k,r+\hat{b}_k} - V^\pi_{P^\star,r} \geqslant -\sqrt{(1-\gamma)^{-1}|\mathcal{A}|\zeta_k}.$$

*Proof.* In this proof, letting $f_k(s,a) = \|\hat{P}_k(\cdot \mid s,a) - P^\star(\cdot \mid s,a)\|_1$, we condition on the event $\forall k, \forall \phi$,

$$\mathbb{E}_{s\sim\rho_k,a\sim U(\mathcal{A})}[f_k^2(s,a)] \leqslant \zeta_k, \mathbb{E}_{s\sim\rho'_k,a\sim U(\mathcal{A})}[f_k^2(s,a)] \leqslant \zeta_k,$$

$$\|\phi(s,a)\|_{\hat{\Sigma}^{-1}_k,\phi} = \Theta(\|\phi(s,a)\|_{\Sigma^{-1}_{\rho_k\times U(\mathcal{A}),\phi}}).$$

From Lemma 14 and Lemma 27, this event happens with probability $1 - \delta$. Then, for any policy $\pi$, from simulation lemma 30,

$$(1-\gamma)(V^\pi_{\hat{P}_k, r+\hat{b}_k} - V^\pi_{P^\star, r})$$

$$= \mathbb{E}_{(s,a)\sim d^\pi_{\hat{P}_k}} \left[ \hat{b}_k(s,a) + \gamma \mathbb{E}_{s'\sim \hat{P}_k(s,a)} \left[ V^\pi_{P^\star, r}(s') \right] - \gamma \mathbb{E}_{s'\sim P^\star(s,a)} \left[ V^\pi_{P^\star, r}(s') \right] \right]$$

$$\gtrsim \mathbb{E}_{(s,a)\sim d^\pi_{\hat{P}_k}} \left[ \min \left( \alpha_k \|\hat{\phi}_k(s,a)\|_{\Sigma^{-1}_{\rho_k \times U(\mathcal{A}), \hat{\phi}_k}}, 2 \right) \right. \tag{11}$$

$$\left. + \gamma \mathbb{E}_{s'\sim \hat{P}_k(s,a)} \left[ V^\pi_{P^\star, r}(s') \right] - \gamma \mathbb{E}_{s'\sim P^\star(s,a)} \left[ V^\pi_{P^\star, r}(s') \right] \right]$$

where in the last step, we replaced the empirical covariance by the population covariance. Note the notation $\lesssim$ is up to universal constants. Here, since $\|V^\pi_{P^\star, r}\|_\infty \leqslant 1$ (since we assume trajectory-wise total reward is normalized between $[0,1]$), we have:

$$|\mathbb{E}_{(s,a)\sim d^\pi_{\hat{P}_k}} \left\{ \mathbb{E}_{s'\sim \hat{P}_k(s,a)} \left[ V^\pi_{P^\star, r}(s') \right] - \mathbb{E}_{s'\sim P^\star(\cdot|s,a)} \left[ V^\pi_{P^\star, r}(s') \right] \right\} | \leqslant \mathbb{E}_{(s,a)\sim d^\pi_{\hat{P}_k}} [f_k(s,a)].$$

The above is further bounded by Lemma 15:

$$|\mathbb{E}_{(s,a)\sim d^\pi_{\hat{P}_k}} [f_k(s,a)]|$$

$$\leqslant \mathbb{E}_{(\tilde{s},\tilde{a})\sim d^\pi_{\hat{P}_k}} \|\hat{\phi}_k(\tilde{s},\tilde{a})\|_{\Sigma^{-1}_{\rho_k \times U(\mathcal{A}), \hat{\phi}_k}} \sqrt{\gamma} \cdot \sqrt{k|\mathcal{A}|\mathbb{E}_{s\sim\rho'_k, a\sim U(\mathcal{A})} [f^2_k(s,a)] + 4\lambda_k d + 4k\zeta_k}$$

$$+ \sqrt{(1-\gamma)|\mathcal{A}|\mathbb{E}_{s\sim\rho_k, a\sim U(\mathcal{A})} [f^2_k(s,a)]}.$$

Then,

$$\mathbb{E}_{(s,a)\sim d^\pi_{\hat{P}_k}} [f_k(s,a)] \lesssim \sqrt{\alpha'_k} \mathbb{E}_{(\tilde{s},\tilde{a})\sim d^\pi_{\hat{P}_k}} \|\hat{\phi}_k(\tilde{s},\tilde{a})\|_{\Sigma^{-1}_{\rho_k \times U(\mathcal{A}), \hat{\phi}_k}} + \sqrt{|\mathcal{A}|\zeta_k(1-\gamma)} \tag{12}$$

where

$$\alpha'_k = \gamma\{k|\mathcal{A}|\zeta_k + \lambda_k d + k\zeta_k\} \lesssim \gamma \left( |\mathcal{A}| + d^2 \right) \ln(|\mathcal{M}|k/\delta).$$

Note we here use $f_k(s,a) \leqslant 2, \mathbb{E}_{s\sim\rho_k, a\sim U(\mathcal{A})}[f_k(s,a)^2] \leqslant \zeta_k$ and $\mathbb{E}_{s\sim\rho'_k, a\sim U(\mathcal{A})}[f_k(s,a)^2] \leqslant \zeta_k$.

Combining all things together,

$$\left| \mathbb{E}_{(s,a)\sim d^\pi_{\hat{P}_k}} \left\{ \mathbb{E}_{s'\sim \hat{P}_k(s,a)} [V^\pi_{P^\star, r}(s')] - \mathbb{E}_{s'\sim P^\star(s,a)} [V^\pi_{P^\star, r}(s')] \right\} \right|$$

$$\leqslant \mathbb{E}_{(s,a)\sim d^\pi_{\hat{P}_k}} [f_k(s,a)]$$

$$\lesssim \sqrt{\alpha'_k} \mathbb{E}_{(\tilde{s},\tilde{a})\sim d^\pi_{\hat{P}_k}} \|\hat{\phi}_k(\tilde{s},\tilde{a})\|_{\Sigma^{-1}_{\rho_k \times U(\mathcal{A}), \hat{\phi}_k}} + \sqrt{(1-\gamma)|\mathcal{A}|\zeta_k}$$

$$\leqslant \alpha_k \mathbb{E}_{(\tilde{s},\tilde{a})\sim d^\pi_{\hat{P}_k}} \|\hat{\phi}_k(\tilde{s},\tilde{a})\|_{\Sigma^{-1}_{\rho_k \times U(\mathcal{A}), \hat{\phi}_k}} + \sqrt{(1-\gamma)|\mathcal{A}|\zeta_k}, \tag{13}$$

where $\alpha_k := \sqrt{\alpha'_k}$.

Going back to (11), we have

$$(1-\gamma)(V^\pi_{\hat{P}_k, r+\hat{b}_k} - V^\pi_{P^\star, r})$$

$$\gtrsim \mathbb{E}_{(s,a)\sim d^\pi_{\hat{P}_k}} \left[ \min \left( \alpha_k \|\hat{\phi}_k(s,a)\|_{\Sigma^{-1}_{\rho_k \times U(\mathcal{A}), \hat{\phi}_k}}, \bar{b} \right) \right.$$

$$\left. + \gamma \mathbb{E}_{s'\sim \hat{P}_k(s,a)} \left[ V^\pi_{P^\star, r}(s') \right] - \gamma \mathbb{E}_{s'\sim P^\star(s,a)} \left[ V^\pi_{P^\star, r}(s') \right] \right]$$

$$\geqslant \mathbb{E}_{(s,a)\sim d^\pi_{\hat{P}_k}} \left[ \min \left( \alpha_k \|\hat{\phi}_k(s,a)\|_{\Sigma^{-1}_{\rho_k \times U(\mathcal{A}), \hat{\phi}_k}}, \bar{b} \right) \right.$$

$$\left. - \min \left( \alpha_k \|\hat{\phi}_k(s,a)\|_{\Sigma^{-1}_{\rho_k \times U(\mathcal{A}), \hat{\phi}_k}} + \sqrt{(1-\gamma)|\mathcal{A}|\zeta_k}, \bar{b} \right) \right]$$

$$\geqslant -\sqrt{(1-\gamma)|\mathcal{A}|\zeta_k}.$$

From the second line to the third line, we again use $\|V^\pi_{P^\star, r}\|_\infty = O(1)$ and 12. This concludes the proof. $\qquad \square$

**Lemma 18.** *Set* $\alpha_k = O(\sqrt{(|\mathcal{A}| + d^2)\,\gamma \ln(|\mathcal{M}|k/\delta)}), \quad \lambda_k = O\left(d \ln(|\mathcal{M}|k/\delta)\right)$ *in algorithm 1. With probability* $1 - \delta$, *we have*

$$\sum_{k=1}^{K}(V_{P^*,r}^{\pi^*}(s_1) - V_{P^*,r}^{\pi_k}(s_1)) + \sum_{k=1}^{K} Y_k(V_{P^*,g}^{\pi^*}(s_1) - V_{P^*,g}^{\pi_k}(s_1))$$

$$\lesssim \sqrt{K \ln\left(1 + \frac{K}{d^2 \ln(|\mathcal{M}|/\delta)}\right) \ln(K|\mathcal{M}|/\delta)} \frac{\xi \bar{b}|\mathcal{A}|d^2}{(1-\gamma)^2}.$$

*Proof.*

$$(V_r^{\pi^*}(s_1) - V_r^{\pi_k}(s_1)) + Y_k(V_g^{\pi^*}(s_1) - V_g^{\pi_k}(s_1))$$

$$\leqslant (V_{\hat{P}_k, r+\hat{b}_k}^{\pi^*}(s_1) - V_r^{\pi_k}(s_1)) + Y_k(V_{\hat{P}_k, g+\hat{b}_k}^{\pi^*}(s_1) - V_g^{\pi_k}(s_1)) + \sqrt{|\mathcal{A}|\zeta_k(1-\gamma)^{-1}}$$

$$\leqslant (V_{\hat{P}_k, r+\hat{b}_k}^{\pi_k}(s_1) - V_r^{\pi_k}(s_1)) + Y_k(V_{\hat{P}_k, g+\hat{b}_k}^{\pi_k}(s_1) - V_g^{\pi_k}(s_1)) + \sqrt{|\mathcal{A}|\zeta_k(1-\gamma)^{-1}}$$

$$= (1-\gamma)^{-1}\mathbb{E}_{(s,a)\sim d_{P^*}^{\pi_k}}\left[\hat{b}_k(s,a) + \gamma\mathbb{E}_{\hat{P}_k(s'|s,a)}[V_{\hat{P}_k, r+\hat{b}_k}^{\pi_k}(s')] - \gamma\mathbb{E}_{P^*(s'|s,a)}\left[V_{\hat{P}_k, r+\hat{b}_k}^{\pi_k}(s')\right]\right]$$

$$+ Y_k(1-\gamma)^{-1}\mathbb{E}_{(s,a)\sim d_{P^*}^{\pi_k}}\left[\hat{b}_k(s,a) + \gamma\mathbb{E}_{\hat{P}_k(s'|s,a)}[V_{\hat{P}_k, g+\hat{b}_k}^{\pi_k}(s')] - \gamma\mathbb{E}_{P^*(s'|s,a)}\left[V_{\hat{P}_k, g+\hat{b}_k}^{\pi_k}(s')\right]\right]$$

$$+ \sqrt{|\mathcal{A}|\zeta_k(1-\gamma)^{-1}}$$

Here we use the 2nd form of Lemma 30 in the last display.

Then, noting $\|\hat{b}_k\|_\infty \leqslant \bar{b}$, $|Y_k| \leqslant \xi$, we have $\|V_{\hat{P}_k, r+\hat{b}_k}^{\pi_k}\|_\infty \leqslant \frac{\bar{b}}{1-\gamma}$, $Y_k\|V_{\hat{P}_k, g+\hat{b}_k}^{\pi_k}\|_\infty \leqslant \frac{\xi\bar{b}}{1-\gamma}$. Combining this fact with the above expansion, we have

$$V_{P^*,r}^{\pi^*} - V_{P^*,r}^{\pi_k} + Y_k(V_{P^*,g}^{\pi^*} - V_{P^*,g}^{\pi_k})$$

$$\leqslant (1-\gamma)^{-1}\underbrace{\mathbb{E}_{(s,a)\sim d_{P^*}^{\pi_k}}[\hat{b}_k(s,a)]}_{(a)} + \left(\frac{\bar{b}}{(1-\gamma)^2}\right)\underbrace{\mathbb{E}_{(s,a)\sim d_{P^*}^{\pi_k}}[f_k(s,a)]}_{(b)}$$

$$+ \xi(1-\gamma)^{-1}\underbrace{\mathbb{E}_{(s,a)\sim d_{P^*}^{\pi_k}}[\hat{b}_k(s,a)]}_{(c)} + \left(\frac{\xi\bar{b}}{(1-\gamma)^2}\right)\underbrace{\mathbb{E}_{(s,a)\sim d_{P^*}^{\pi_k}}[f_k(s,a)]}_{(d)} + \sqrt{|\mathcal{A}|\zeta_k(1-\gamma)^{-1}}.$$

$$(14)$$

First, we calculate the first term (a) in Inequality 14. Following Lemma 16 and noting the bonus $\hat{b}_k$ is $O(1)$, we have

$$\mathbb{E}_{(s,a)\sim d_{P^*}^{\pi_k}}\left[\hat{b}_k(s,a)\right]$$

$$\lesssim \mathbb{E}_{(s,a)\sim d_{P^*}^{\pi_k}}\left[\min\left(\alpha_k\|\hat{\phi}_k(s,a)\|_{\Sigma_{\rho_k \times \mathcal{U}(\mathcal{A}), \hat{\phi}_k}^{-1}}, \bar{b}\right)\right]$$

$$\lesssim \mathbb{E}_{(\tilde{s},\tilde{a})\sim d_{P^*}^{\pi_k}}\|\phi^*(\tilde{s},\tilde{a})\|_{\Sigma_{\rho_k, \phi^*}^{-1}} \cdot$$

$$\sqrt{k\gamma|\mathcal{A}|\alpha_k^2 \mathbb{E}_{s\sim\rho_k, a\sim\mathcal{U}(\mathcal{A})}\left[\|\hat{\phi}_k(s,a)\|_{\Sigma_{\rho_k \times \mathcal{U}(\mathcal{A}), \hat{\phi}_k}^{-1}}^2\right] + d\gamma\lambda_k\bar{b}^2}$$

$$+ \sqrt{|\mathcal{A}|\alpha_k^2 \mathbb{E}_{s\sim\rho_k, a\sim\mathcal{U}(\mathcal{A})}\left[\|\hat{\phi}_k(s,a)\|_{\Sigma_{\rho_k \times \mathcal{U}(\mathcal{A}), \hat{\phi}_k}^{-1}}^2\right]}(1-\gamma).$$

Note that we use the fact that $B = \bar{b}$ when applying Lemma 16. In addition, we have

$$k\mathbb{E}_{s\sim\rho_k, a\sim U(\mathcal{A})}\left[\|\hat{\phi}_k(s,a)\|_{\Sigma_{\rho_k \times U(\mathcal{A}), \hat{\phi}_k}^{-1}}^2\right]$$

$$= k\text{Tr}(\mathbb{E}_{\rho_k \times U(\mathcal{A})}[\hat{\phi}_k\hat{\phi}_k^\top]\{k\mathbb{E}_{\rho_k \times U(\mathcal{A})}[\hat{\phi}_k\hat{\phi}_k^\top] + \lambda_k I\}^{-1})$$

$$\leqslant d.$$

Then,

$$
\mathbb{E}_{(s,a)\sim d_{P^\star}^{\pi_k}}\left[\hat{b}_k(s,a)\right]
$$

$$
\leqslant \mathbb{E}_{(\tilde{s},\tilde{a})\sim d_{P^\star}^{\pi_k}}\|\phi^\star(\tilde{s},\tilde{a})\|_{\Sigma_{\rho_k,\phi^\star}^{-1}}\sqrt{\gamma d|\mathcal{A}|\alpha_k^2+\gamma d\lambda_k\bar{b}^2}+\sqrt{d|\mathcal{A}|\alpha_k^2(1-\gamma)/k}.
$$

Second, we calculate the term (b) in inequality 14. Following Lemma 16 and noting $f_k^2(s,a)$ is upper-bounded by $4$ (i.e., $B=4$ in Lemma 16), we have

$$
\mathbb{E}_{(s,a)\sim d_{P^\star}^{\pi_k}}[f_k(s,a)]
$$

$$
\leqslant \mathbb{E}_{(\tilde{s},\tilde{a})\sim d_{P^\star}^{\pi_k}}\|\phi^\star(\tilde{s},\tilde{a})\|_{\Sigma_{\rho_k,\phi^\star}^{-1}}\sqrt{k|\mathcal{A}|\gamma\mathbb{E}_{s\sim\rho_k,a\sim U(\mathcal{A})}\left[f_k^2(s,a)\right]+4\gamma\lambda_k d}
$$

$$
+\sqrt{|\mathcal{A}|\mathbb{E}_{s\sim\rho_k,a\sim U(\mathcal{A})}\left[f_k^2(s,a)(1-\gamma)\right]}
$$

$$
\leqslant \mathbb{E}_{(\tilde{s},\tilde{a})\sim d_{P^\star}^{\pi_k}}\|\phi^\star(\tilde{s},\tilde{a})\|_{\Sigma_{\rho_k,\phi^\star}^{-1}}\sqrt{k|\mathcal{A}|\gamma\zeta_k+4\gamma\lambda_k d}+\sqrt{|\mathcal{A}|\zeta_k(1-\gamma)}
$$

$$
\leqslant \mathbb{E}_{(\tilde{s},\tilde{a})\sim d_{P^\star}^{\pi_k}}\|\phi^\star(\tilde{s},\tilde{a})\|_{\Sigma_{\rho_k,\phi^\star}^{-1}}\alpha_k+\sqrt{|\mathcal{A}|\zeta_k(1-\gamma)},
$$

where in the second inequality, we use $\mathbb{E}_{s\sim\rho_k,a\sim U(\mathcal{A})}[f_k^2(s,a)]\leqslant\zeta_k$, and in the last line, recall $\sqrt{\gamma}\sqrt{k|\mathcal{A}|\zeta_k+\lambda_k d+k\zeta_k}\lesssim\alpha_k$.

Similarly, we can upper bound the term (c) and (d). Combining the calculation, we have:

$$
V_{P^\star,r}^{\pi^\star}-V_{P^\star,r}^{\pi_k}+Y_k(V_{P^\star,g}^{\pi^\star}-V_{P^\star,g}^{\pi_k})
$$

$$
\lesssim \frac{1+\xi}{1-\gamma}\left(\mathbb{E}_{(\tilde{s},\tilde{a})\sim d_{P^\star}^{\pi_k}}\|\phi^\star(\tilde{s},\tilde{a})\|_{\Sigma_{\rho_k,\phi^\star}^{-1}}\sqrt{d|\mathcal{A}|\alpha_k^2+d\lambda_k\bar{b}^2}+\sqrt{\frac{d|\mathcal{A}|\alpha_k^2(1-\gamma)}{k}}\right)
$$

$$
+\frac{(1+\xi)\bar{b}}{(1-\gamma)^2}\left(\mathbb{E}_{(\tilde{s},\tilde{a})\sim d_{P^\star}^{\pi_k}}\|\phi^\star(\tilde{s},\tilde{a})\|_{\Sigma_{\rho_k,\phi^\star}^{-1}}\alpha_k+\sqrt{|\mathcal{A}|\zeta_k(1-\gamma)}\right).
$$

Hereafter, we take the dominating term out. First, recall

$$
\alpha_k\lesssim\sqrt{(|\mathcal{A}|+d^2)\ln(K|\mathcal{M}|/\delta)}\lesssim\sqrt{|\mathcal{A}|d^2\ln(K|\mathcal{M}|/\delta)}.
$$

Then, denote $D(\Phi)=\ln\det(\sum_{k=1}^K\mathbb{E}_{(\tilde{s},\tilde{a})\sim d_{P^\star}^{\pi_k}}[\phi^\star(\tilde{s},\tilde{a})\phi^\star(\tilde{s},\tilde{a})^\top])-\ln\det(\lambda_1 I)$, we have

$$
\sum_{k=1}^K\mathbb{E}_{(\tilde{s},\tilde{a})\sim d_{P^\star}^{\pi_k}}\|\phi^\star(\tilde{s},\tilde{a})\|_{\Sigma_{\rho_k,\phi^\star}^{-1}}
$$

$$
\leqslant\sqrt{K\sum_{k=1}^K\mathbb{E}_{(\tilde{s},\tilde{a})\sim d_{P^\star}^{\pi_k}}[\phi^\star(\tilde{s},\tilde{a})^\top\Sigma_{\rho_k,\phi^\star}^{-1}\phi^\star(\tilde{s},\tilde{a})]}\qquad\text{(CS inequality)}
$$

$$
\lesssim\sqrt{K\cdot D(\Phi)}\qquad\qquad\text{(Lemma 28 and }\lambda_1\leqslant\cdots\leqslant\lambda_K)
$$

$$
\leqslant\sqrt{dK\ln\left(1+\frac{K}{d\lambda_1}\right)}.
$$

The last inequality uses potential function bound and Lemma 29, noting $\|\phi^\star(s,a)\|_2\leqslant 1$ for any $(s,a)$.

Finally,

$$\sum_{k=1}^{K} V_{P^\star,r}^{\pi} - V_{P^\star,r}^{\pi_k} + Y_k(V_{P^\star,g}^{\pi} - V_{P^\star,g}^{\pi_k})$$

$$\lesssim \frac{1+\xi}{1-\gamma}\left(\sqrt{dK\ln\left(1+\frac{K}{d\lambda_1}\right)}\sqrt{d|\mathcal{A}|\alpha_K^2 + d\lambda_K\bar{b}^2} + \sum_{k=1}^{K}\sqrt{\frac{d|\mathcal{A}|\alpha_k^2(1-\gamma)}{k}}\right)$$

$$+ \frac{(1+\xi)\bar{b}}{(1-\gamma)^2}\left(\sqrt{dK\ln\left(1+\frac{K}{d\lambda_1}\right)}\alpha_K + \sum_{k=1}^{K}\sqrt{|\mathcal{A}|\zeta_k(1-\gamma)}\right)$$

$$\lesssim \frac{\xi\bar{b}}{1-\gamma}\sqrt{dK\ln\left(1+\frac{K}{d\lambda_1}\right)}\sqrt{d|\mathcal{A}|\alpha_K^2} + \frac{\xi\bar{b}}{(1-\gamma)^2}\sqrt{dK\ln\left(1+\frac{K}{d\lambda_1}\right)}\alpha_K$$

(Some algebra. We take the dominating term out.)

$$\lesssim \sqrt{dK\ln\left(1+\frac{K}{d\lambda_1}\right)}\frac{\xi\bar{b}|\mathcal{A}|d^{3/2}\ln^{1/2}(K|\mathcal{M}|/\delta)}{(1-\gamma)^2}.$$

This concludes the proof.

$\square$

Then, we have the regret and soft constraint violation guarantee.

**Theorem 19** (Regret and Soft Constraint Violation of Algorithm 1). *Set* $\alpha_k = O(\sqrt{(|\mathcal{A}| + d^2)\gamma\ln(|\mathcal{M}|k/\delta)})$, $\lambda_k = O(d\ln(|\mathcal{M}|k/\delta))$, $\eta = \frac{2(1-\gamma)}{\theta\bar{b}\sqrt{K}}$ *in algorithm 1.* *With probability* $1-\delta$, *we have*

$$Regret(K) \leqslant \mathcal{O}\left(\frac{|\mathcal{A}|\bar{b}d^2\sqrt{K}}{\theta(1-\gamma)^2}\right), Violation_{soft}(K) \leqslant \mathcal{O}\left(\frac{|\mathcal{A}|\bar{b}d^2\sqrt{K}}{\theta(1-\gamma)^2}\right).$$

*Proof.* We, first, show the regret bound. Note from Lemma 13, Lemma 17 and Lemma 18, for $Y \in [0,\xi]$, with probability $1-\delta$, we have

$$\sum_{k=1}^{K}(V_{P^*,r}^{\pi^*}(s_1) - V_{P^*,r}^{\pi_k}(s_1)) + Y(b - V_{P^*,g}^{\pi_k}(s_1))$$

$$\lesssim \frac{Y^2}{2\eta} + \frac{\eta\bar{b}^2 K}{2(1-\gamma)^2} + \xi\sqrt{\frac{K|\mathcal{A}|\ln(K|\mathcal{M}|/\delta)}{(1-\gamma)}} + \sqrt{K\ln\left(1+\frac{K}{d^2\ln(|\mathcal{M}|/\delta)}\right)\ln(K|\mathcal{M}|/\delta)}\frac{\xi\bar{b}|\mathcal{A}|d^2}{(1-\gamma)^2}.$$

(15)

Replacing $Y$ with 0 in (15), we have

$$\sum_{k=1}^{K}(V_{P^*,r}^{\pi^*}(s_1) - V_{P^*,r}^{\pi_k}(s_1))$$

$$\lesssim \frac{\eta\bar{b}^2 K}{2(1-\gamma)^2} + \xi\sqrt{\frac{K|\mathcal{A}|\ln(K|\mathcal{M}|/\delta)}{(1-\gamma)}} + \sqrt{K\ln\left(1+\frac{K}{d^2\ln(|\mathcal{M}|/\delta)}\right)\ln(K|\mathcal{M}|/\delta)}\frac{\xi\bar{b}|\mathcal{A}|d^2}{(1-\gamma)^2}.$$

Substituting $\eta = \frac{2(1-\gamma)}{\theta\bar{b}\sqrt{K}}$ and $\xi = \frac{2}{\theta}$ into the equation, we have the result.

We, now, show the violation bound. Since

$$\sum_{k=1}^{K}(V_{P^*,r}^{\pi^*}(s_1) - V_{P^*,r}^{\pi_k}(s_1)) + Y(b - V_{P^*,g}^{\pi_k}(s_1))$$

$$\lesssim \frac{Y^2}{2\eta} + \frac{\eta \bar{b}^2 K}{2(1-\gamma)^2} + \xi\sqrt{\frac{K|\mathcal{A}|\ln(K|\mathcal{M}|/\delta)}{(1-\gamma)}}$$

$$+ \sqrt{K\ln\left(1 + \frac{K}{d^2\ln(|\mathcal{M}|/\delta)}\right)\ln(K|\mathcal{M}|/\delta)}\frac{\xi\bar{b}|\mathcal{A}|d^2}{(1-\gamma)^2}.$$

Put $\eta = \frac{2(1-\gamma)}{\theta\bar{b}\sqrt{K}}$, $\xi = \frac{2}{\theta}$ and $Y \leqslant \xi$, we have

$$\sum_{k=1}^{K}(V_{P^*,r}^{\pi^*}(s_1) - V_{P^*,r}^{\pi_k}(s_1)) + Y(b - V_{P^*,g}^{\pi_k}(s_1))$$

$$\lesssim \frac{2\bar{b}\sqrt{K}}{\theta(1-\gamma)} + \sqrt{\frac{K|\mathcal{A}|\ln(K|\mathcal{M}|/\delta)}{(1-\gamma)^2\theta}} + \sqrt{K\ln\left(1 + \frac{K}{d^2\ln(|\mathcal{M}|/\delta)}\right)\ln(K|\mathcal{M}|/\delta)}\frac{\xi\bar{b}|\mathcal{A}|d^2}{(1-\gamma)^2}.$$

Now, there exists a policy $\pi'$ such that $V_r^{\pi'} = \frac{1}{K}\sum_{k=1}^{K}V_r^{\pi_k}$, $V_g^{\pi'} = \frac{1}{K}\sum_{k=1}^{K}V_g^{\pi_k}$. By the occupancy measure, $V_r^{\pi}$ and $V_g^{\pi}$ are linear in occupancy measure induced by $\pi$. Thus, the average of $K$ occupancy measure also produces an occupancy measure which induces policy $\pi'$ and $V_r^{\pi'}$, and $V_g^{\pi'}$. We take $Y = 0$ when $\sum_{k=1}^{K}(b - V_g^{\pi_k}(s_1^k)) < 0$, otherwise $Y = \xi$. Hence, we have

$$V_{P^*,r}^{\pi^*}(s_1) - \frac{1}{K}\sum_{k=1}^{K}V_{P^*,r}^{\pi_k}(s_1) + \xi(b - \frac{1}{K}\sum_{k=1}^{K}V_{P^*,g}^{\pi_k}(s_1))_+$$

$$= V_{P^*,r}^{\pi^*}(s_1) - V_{P^*,r}^{\pi'}(x_1) + \xi[b - V_{P^*,g}^{\pi'}(s_1)]_+$$

$$\leqslant \frac{2}{\sqrt{K}\theta(1-\gamma)^2} + \sqrt{\frac{|\mathcal{A}|\ln(K|\mathcal{M}|/\delta)}{(1-\gamma)^2\theta K}} + \sqrt{\frac{1}{K}\ln\left(1 + \frac{K}{d^2\ln(|\mathcal{M}|/\delta)}\right)\ln(K|\mathcal{M}|/\delta)}\frac{\xi\bar{b}|\mathcal{A}|d^2}{(1-\gamma)^2}.$$

Since $\xi = \frac{2}{\theta}$, and using the result of strong duality (Lemma 31), we have

$$(b - \frac{1}{K}\sum_{k=1}^{K}V_{P^*,g}^{\pi_k}(s_1^k))_+ \leqslant \mathcal{O}(\frac{\bar{b}|\mathcal{A}|d^2}{\theta(1-\gamma)^2\sqrt{K}}) \tag{16}$$

Hence, the result follows. □

**Corollary 20** (PAC Guarantee of Algorithm 1). *After interacting with the environments for $K = \mathcal{O}(\frac{\epsilon^2\theta^2(1-\gamma)^4}{|\mathcal{A}|\bar{b}^2d^4\ln(|\mathcal{M}|/\delta)})$ episodes, we have $\frac{1}{K}\sum_{k=1}^{K}V_{P^*,r}^{\pi^*}(s_1) - V_{P^*,r}^{\pi_k}(s_1) \leqslant \epsilon$ with high probability.*

*Proof.* It directly follows from the standard regret to PAC reduction (21). From Theorem 19, when K is

$$\mathcal{O}(\frac{\epsilon^2\theta^2(1-\gamma)^4}{|\mathcal{A}|\bar{b}^2d^4\ln(|\mathcal{M}|/\delta)})$$

with probability $1 - \delta$, we can ensure $\frac{1}{K}\sum_{k=1}^{K}V_{P^*,r}^{\pi^*}(s_1) - V_{P^*,r}^{\pi_k}(s_1) \leqslant \epsilon$. □

## C  PROOF OF THEOREM 11

This section provides detailed proofs for our results on hard constraint violation.

**Assumption 3** (positive action-gap). *For all $\hat{P}_k$, we assume that there exists a minimum action-gap $\delta_{min}$ such that:*

$$\delta_{min} = \min_s \min_{a \neq a^*(s)}\left(Q_{\hat{P}_k}^{\pi^*}(s, a^*(s) - Q_{\hat{P}_k}^{\pi^*}(s, a)\right) > 0.$$

Assumption 3 ensures that the optimal action at each state is clearly separated from the suboptimal actions, which is standard for softmax-policy analyses (50). All results in this section are derived under this assumption.

In analogy to the soft constraint setting and in conjunction with the properties of the soft-max policy, we present the following lemma.

**Lemma 21.** *Set* $\alpha_k = O(\sqrt{(|\mathcal{A}| + d^2)\gamma \ln(|\mathcal{M}|k/\delta)})$, $\lambda_k = O\left(d\ln(|\mathcal{M}|k/\delta)\right)$, $\tau = \frac{\sqrt{K}}{(1-\gamma)^3}$, $\eta = \frac{1}{\sqrt{K}(1-\gamma)}$ *in algorithm 2. With probability* $1 - \delta$, *we have*

$$\sum_{k=1}^{K}(V_r^{\pi^*}(s_1) - V_r^{\pi_k}(s_1)) + \sum_{k=1}^{K}Y_k(V_g^{\pi^*}(s_1) - V_g^{\pi_k}(s_1))$$

$$\lesssim \sqrt{\ln\left(1 + \frac{K}{d^2\ln(|\mathcal{M}|/\delta)}\right)\ln(K|\mathcal{M}|/\delta)}\frac{K^{3/4}|\mathcal{A}|\bar{b}d^2}{(1-\gamma)^2} + \frac{|\mathcal{A}|\exp(-\frac{\delta_{\min}}{\tau})}{(1-\gamma)^2}.$$

To upper bound the regret, we decompose it into two main components:

$$Regret(K) \leqslant \underbrace{\sum_{k=1}^{K}(V_r^{\pi^*}(s_1) + Y_k V_g^{\pi^*}(s_1) - V_r^{\pi_k}(s_1) - Y_k V_g^{\pi_k}(s_1))}_{\mathcal{T}_1} + \underbrace{\sum_{k=1}^{K}Y_k(V_g^{\pi_k}(s_1) - b)}_{\mathcal{T}_2}$$

(17)

Using Lemma 21, we can bound $\mathcal{T}_1$.

To bound $\mathcal{T}_2$, we first define the set of episodes where the constraint is violated:

$$\Gamma = \{k : (V_g^{\pi_k}(s_1) < b\}.$$

For episodes in $\Gamma$, $\mathcal{T}_2$ is upper bounded by 0. We only need to consider the episodes in the complement set $\Gamma^C$.

Let $V_g^{\pi_k, Y}$ be the estimated value function computed by the Algorithm 2 when the dual variable is $Y$. Before upper bound $\mathcal{T}_2$ for episodes in the set $\Gamma^C$, we show that due to the softmax property, the difference between $V_g^{\pi_k, Y}$ and $V_g^{\pi_k, Y+\eta}$ is bounded.

**Lemma 22.** *Denote* $\pi_k^Y$ *as the soft-max policy computed at iteration $k$ with $Y$. Then,* $|V_g^{\pi_k, Y}(s_1) - V_g^{\pi_k, Y+\eta}(s_1)| \leqslant \mathcal{O}(\frac{\eta}{\tau(1-\gamma)})$.

*Proof.* For any state $s$, the value function satisfies the Bellman equation:

$$V_g^{\pi_k^Y} = \mathbb{E}_{a\sim\pi_k^Y(s)}\left[g(s,a) + \gamma\mathbb{E}_{s'\sim P^\star}[V_g^{\pi_k^Y}(s')]\right]$$

$$V_g^{\pi_k^{Y'}} = \mathbb{E}_{a\sim\pi_k^{Y'}(s)}\left[g(s,a) + \gamma\mathbb{E}_{s'\sim P^\star}[V_g^{\pi_k^{Y'}}(s')]\right]$$

Let $\Delta_g = \sup_s |V_g^{\pi_k^Y}(s) - V_g^{\pi_k^{Y'}}(s)|$. Subtracting the two equations, we have:

$$\left|V_g^{\pi_k^Y}(s) - V_g^{\pi_k^{Y'}}(s)\right|$$

$$= |\mathbb{E}_{a\sim\pi_k^Y(\cdot|s)}\left[g(s,a) + \gamma\mathbb{E}_{s'\sim P^\star}[V_g^{\pi_k^Y}(s')]\right] - \mathbb{E}_{a\sim\pi_k^{Y'}(\cdot|s)}\left[g(s,a) + \gamma\mathbb{E}_{s'\sim P^\star}[V_g^{\pi_k^{Y'}}(s')]\right]|$$

$$\leqslant \underbrace{\left|\mathbb{E}_{a\sim\pi_k^Y(\cdot|s)}\left[\gamma\mathbb{E}_{s'\sim P^\star}[V_g^{\pi_k^Y}(s') - V_g^{\pi_k^{Y'}}(s')]\right]\right|}_{\Gamma_1} + \underbrace{\left|\mathbb{E}_{a\sim\pi_k^Y(\cdot|s)}\left[g(s,a) + \gamma\mathbb{E}_{s'\sim P^\star}[V_g^{\pi_k^{Y'}}(s')]\right]\right.}_{\Gamma_2}$$

$$\underbrace{\left.- \mathbb{E}_{a\sim\pi_k^{Y'}(\cdot|s)}\left[g(s,a) + \gamma\mathbb{E}_{s'\sim P^\star}[V_g^{\pi_k^{Y'}}(s')]\right]\right|}_{\Gamma_2}$$

For term $\Gamma_1$,

$$\Gamma_1 \leqslant \gamma \mathbb{E}_{a,s'} \left[ V_g^{\pi_k^Y}(s') - V_g^{\pi_k^{Y'}}(s') \right] \leqslant \gamma \sup_{s'} |V_g^{\pi_k^Y}(s') - V_g^{\pi_k^{Y'}}(s')| = \gamma \Delta_g.$$

For term $\Gamma_2$, by the update rule of the policy, we have:

$$\Gamma_2 = \left| \sum_a \left( \pi_k^Y(a|s) - \pi_k^{Y'}(a|s) \right) \cdot \left( g(s,a) + \gamma \mathbb{E}_{s'}[V_g^{\pi_k^{Y'}}(s')] \right) \right|$$

$$\leqslant \|\pi_k^Y(\cdot|s) - \pi_k^{Y'}(\cdot|s)\|_1 \cdot \|g + \gamma P^* V_g^{\pi_k^{Y'}}\|_\infty$$

$$\leqslant \|\pi_k^Y - \pi_k^{Y'}\|_1.$$

By the property of the soft-max (Theorem 4.4 in (51))

$$\Gamma_2 \leqslant \frac{2\eta}{\tau}.$$

Substituting $\Gamma_1$ and $\Gamma_2$:

$$\left| V_g^{\pi_k^Y}(s) - V_g^{\pi_k^{Y'}}(s) \right| \leqslant \gamma \Delta_g + \frac{2\eta}{\tau}.$$

Take the supremum over all states, we have:

$$\Delta_g \leqslant \gamma \Delta_g + \frac{2\eta}{\tau}.$$

Solving for it gives:

$$\Delta_g \leqslant \frac{2\eta}{\tau(1-\gamma)}.$$

Thus, the difference between the value functions is bounded as claimed. $\qquad\square$

**Lemma 23.** *Set* $\tau = \frac{\sqrt{K}}{(1-\gamma)^3}$, $\eta = \frac{1}{\sqrt{K}(1-\gamma)}$. *For any* $k \in \Gamma^C$, $Y_k(V_g^{\pi_k,Y_k}(s_1) - b) \leqslant \mathcal{O}(\frac{1}{(1-\gamma)^2 K})$.

*Proof.* If $Y_k = 0$, ,the result is trivially true. When $Y_k > 0$ and $k \in \Gamma^C$, we know that $V_g^{\pi_k,Y_k}(s_1) \geqslant b$ while $V_g^{\pi_k^{Y_k-\eta},Y_k-\eta}(s_1) < b$. From Lemma 22,

$$|V_g^{\pi_k,Y_k}(s_1) - V_g^{\pi_k^{Y_k-\eta},Y_k-\eta}(s_1)| \leqslant \mathcal{O}(\frac{\eta}{K(1-\gamma)})$$

$$V_g^{\pi_k,Y_k}(s_1) \leqslant V_g^{\pi_k^{Y_k-\eta},Y_k-\eta}(s_1) + \mathcal{O}(\frac{\eta}{K(1-\gamma)}) \leqslant b + \mathcal{O}(\frac{\eta}{K(1-\gamma)})$$

Since $Y_k \leqslant \sqrt{K}$ and $\eta = \frac{1}{\sqrt{K}(1-\gamma)}$,

$$Y_k(V_g^{\pi_k,Y_k}(s_1) - b) \leqslant \mathcal{O}(\frac{1}{(1-\gamma)^2 K}). \tag{18}$$

$$\square$$

Then, we can derive the final bounds on regret and hard constraint violation for Algorithm 2.

**Theorem 24** (Regret and Hard Constraint Violation of Algorithm 2). *Set* $\alpha_k = O(\sqrt{(|\mathcal{A}| + d^2)\gamma \ln(|\mathcal{M}|k/\delta)})$, $\lambda_k = O(d\ln(|\mathcal{M}|k/\delta))$, $\tau = \frac{\sqrt{K}}{(1-\gamma)^3}$, $\eta = \frac{1}{\sqrt{K}(1-\gamma)}$ *in algorithm 2. With probability* $1 - \delta$, *we have*

$$Regret(K) \leqslant \mathcal{O}\left( \frac{|\mathcal{A}|\bar{b}d^2 K^{3/4}}{(1-\gamma)^2} \right), Violation_{hard}(K) \leqslant \mathcal{O}\left( \frac{|\mathcal{A}|\bar{b}d^2 \sqrt{K}}{(1-\gamma)^2} \right)$$

*Proof.* We, first, show the regret bound. From (17), Lemma 23 and Lemma 21, for $Y \in [0, \sqrt{K}]$, with probability $1 - \delta$, we have

$$Regret(K) \lesssim \sqrt{\ln\left(1 + \frac{K}{d^2 \ln(|\mathcal{M}|/\delta)}\right) \ln(K|\mathcal{M}|/\delta)} \frac{K^{3/4}|\mathcal{A}|\bar{b}d^2}{(1-\gamma)^2} + \frac{|\mathcal{A}|\exp(-\frac{\delta_{\min}}{\tau})}{(1-\gamma)^2} + \frac{1}{(1-\gamma)^2}.$$

We, now, show the violation bound. Since

$$\sum_{k=1}^{K}(V_r^{\pi^*}(s_1) - V_r^{\pi_k}(s_1)) + \sum_{k=1}^{K} Y_k(V_g^{\pi^*}(s_1) - V_g^{\pi_k}(s_1))$$

$$\lesssim \sqrt{\ln\left(1 + \frac{K}{d^2 \ln(|\mathcal{M}|/\delta)}\right) \ln(K|\mathcal{M}|/\delta)} \frac{K^{3/4}|\mathcal{A}|\bar{b}d^2}{(1-\gamma)^2}.$$

We have

$$\sum_{k=1}^{K} Y_k(b - V_g^{\pi_k}(s_1)) \leqslant \sum_{k=1}^{K} Y_k(V_g^{\pi^*}(s_1) - V_g^{\pi_k}(s_1))$$

$$\lesssim \sqrt{\ln\left(1 + \frac{K}{d^2 \ln(|\mathcal{M}|/\delta)}\right) \ln(K|\mathcal{M}|/\delta)} \frac{K^{3/4}|\mathcal{A}|\bar{b}d^2}{(1-\gamma)^2}.$$

Note that by the update of $Y_k$, we have $Y_k = \sqrt{K}$ for all $k$ such that $b - V_g^{\pi_k}(s_1) > 0$. Since $\sum_{k=1}^{K}(b - V_g^{\pi_k}(s_1)_+ = \sum_{k \in \Gamma}(b - V_g^{\pi_k})_+$, we have that

$$\sum_{k=1}^{K}(b - V_g^{\pi_k}(s_1))_+ = \sum_{k \in \Gamma}(b - V_g^{\pi_k}(s_1))_+ \lesssim \sqrt{K \ln\left(1 + \frac{K}{d^2 \ln(|\mathcal{M}|/\delta)}\right) \ln(K|\mathcal{M}|/\delta)} \frac{|\mathcal{A}|\bar{b}d^2}{(1-\gamma)^2}.$$

Hence, the result follows. $\square$

## D ANALYSIS FOR ZERO SOFT CONSTRAINT VIOLATION

To achieve zero soft constraint violation, we reformulate the original constrained problem as a tighter optimization problem–

$$\text{maximize}_{\pi \in \Delta(\mathcal{A}|\mathcal{S})} V_r^{\pi}(s_1) \quad \text{subject to } V_g^{\pi}(s_1) \geqslant b + \zeta. \tag{19}$$

Instead of enforcing the original threshold $b$, we tighten the constraint to $b + \zeta$, where $\zeta > 0$ is carefully chosen. By ensuring $\zeta \leqslant \theta/2$, Slater's condition remains satisfied, preserving strong duality (47). This allows us to derive an optimal dual variable $Y^{\zeta}$ for the tighter problem, bounded as:

$$Y^{\zeta} \leqslant \frac{V^{\pi^{\zeta,*}}(s_1) - V_r^{\bar{\pi}}(s_1)}{b + \gamma - (b + \zeta)} \leqslant \frac{4}{\theta(1-\gamma)}. \tag{20}$$

Then, we state the formal guarantee for the tighter problem.

**Theorem 25.** *In Algorithm 1, replacing $b = b + \zeta$, and set $\xi = \frac{4}{(1-\gamma)\theta}$. Then, with probability at least $1 - p$, we have*

$$\text{Regret}(K) \leqslant \mathcal{O}\left(\frac{|\mathcal{A}|\bar{b}d^2\sqrt{K}}{\theta(1-\gamma)^2}\right) + K\frac{\zeta}{\theta(1-\gamma)}$$

$$\text{Violation}(K) \leqslant \max\left\{\mathcal{O}\left(\frac{|\mathcal{A}|\bar{b}d^2\sqrt{K}}{\theta(1-\gamma)^2}\right) - K\zeta, 0\right\}, \tag{21}$$

*where $\zeta = \min\{\mathcal{O}\left(\frac{|\mathcal{A}|\bar{b}d^2\sqrt{K}}{K\theta(1-\gamma)^2}\right), \theta/2\}$.*

When $\mathcal{O}\left(\frac{|\mathcal{A}|\bar{b}d^2\sqrt{K}}{K\theta(1-\gamma)^2}\right) \leqslant \theta/2$, the violation term becomes non-positive, ensuring zero soft constraint violation for sufficiently large $K$. Plugging in the upper bound for $\zeta$, we obtain the upper bound on regret as

$$\text{Regret}(K) \leqslant \mathcal{O}\left(\frac{|\mathcal{A}|\bar{b}d^2\sqrt{K}}{\theta(1-\gamma)^2}\right) + \mathcal{O}\left(\frac{|\mathcal{A}|\bar{b}d^2\sqrt{K}}{\theta^2(1-\gamma)^3}\right)$$

where we replace the upper bound of $\zeta$ by $\mathcal{O}\left(\frac{|\mathcal{A}|\bar{b}d^2\sqrt{K}}{K\theta(1-\gamma)^3}\right)$. Thus, the upper bound on regret is $\mathcal{O}\left(\frac{|\mathcal{A}|\bar{b}d^2\sqrt{K}}{\theta^2(1-\gamma)^3}\right)$.

Before proving Theorem 25, we first introduce the following lemma:

**Lemma 26.** *If $\pi^{\zeta,*}$ is the optimal solution of (19), then*

$$V_r^{\pi^*}(x_1) - V_r^{\pi^{\zeta,*}}(x_1) \leqslant \frac{\zeta}{\theta(1-\gamma)}. \tag{22}$$

The proof of this lemma for finite state MDPs can be found in (52); we extend it to low-rank MDPs.

*Proof.* Denote $\nu^\pi = d_{P*}^\pi$. Let $\nu^*(s,a)$ corresponds to the state-action occupancy measure for the optimal policy $\pi^*$ and denote $\nu^\zeta(s,a) = (1-\zeta/\gamma)\nu^*(s,a) + \zeta/\gamma\nu^{\bar{\pi}}(s,a)$, where $\bar{\pi}$ is a policy such that $V_g^{\bar{\pi}}(s_1) \geqslant b + \theta$.

We have

$$\int_{s,a} g(s,a)d\nu^\zeta(s,a) \geqslant (1-\zeta/\gamma)b + \zeta/\gamma(b+\gamma) = b + \zeta \tag{23}$$

Hence, the state-action occupancy measure $\nu^\zeta(s,a)$ is feasible for the tighter CMDP. Now, we have

$$\int_{s,a} r(s,a)d\nu^\zeta(s,a) \tag{24}$$

$$= (1-\zeta/\theta)\int_{s,a} r(s,a)d\nu^*(s,a) + \zeta/\theta\int_{s,a} r(s,a)d\nu^{\bar{\pi}}(s,a)$$

$$\geqslant (1-\zeta/\theta)V_r^*(s_1)$$

Since $\nu^\zeta(s,a)$ is feasible, then $V_r^{\pi^{\zeta,*}}(s_1) \geqslant \int_{s,a} r(s,a)d\nu^\zeta(s,a)$. Thus,

$$V_r^*(s_1) - V_r^{\zeta,*}(s_1) \leqslant \zeta/\theta V_r^*(s_1) \leqslant \frac{\zeta}{\theta(1-\gamma)} \tag{25}$$

Hence, the result follows. $\square$

Now, we are able to proof Theorem 25.

*Proof of Theorem 25.* First, we prove the upper bound on regret. The regret can be decomposed as the following:

$$\text{Regret}(K) = \sum_{k=1}^{K}(V_r^{\pi^*}(s_1) - V^{\pi^{\zeta,*}}(s_1)) + \sum_{k=1}^{K}(V_r^{\pi^{\zeta,*}}(s_1) - V^{\pi_k}(s_1)) \tag{26}$$

The first term can be bounded via Lemma 26.

Since the tighter optimization problem is also CMDP, we note that the second term in the right hand side of (26) is essentially the regret of the tighter CMDP.

Hence, from Theorem 6 and Lemma 26 we obtain the expression of the regret bound in Theorem 25.

**Constraint Violation**: Again applying Theorem 6 to the tighter optimization problem (19), we obtain

$$\sum_{k=1}^{K} (b + \zeta - V_g^{\pi_k}(s_1))_+ \leqslant \mathcal{O}\left(\frac{|\mathcal{A}|\bar{b}d^2\sqrt{K}}{\theta(1-\gamma)^2}\right).$$

Hence,

$$\sum_{k=1}^{K}(b + \zeta - V_g^{\pi_k}(s_1)) \leqslant \sum_{k=1}^{K}(b + \zeta - V_g^{\pi_k}(s_1))_+ \leqslant \mathcal{O}\left(\frac{|\mathcal{A}|\bar{b}d^2\sqrt{K}}{\theta(1-\gamma)^2}\right).$$

Rearranging yields:

$$\sum_{k=1}^{K}(b - V_g^{\pi_k}(s_1)) \leqslant \mathcal{O}\left(\frac{|\mathcal{A}|\bar{b}d^2\sqrt{K}}{\theta(1-\gamma)^2}\right) - K\zeta.$$

Hence, we have

$$\left[\sum_{k=1}^{K} b - V_g^{\pi_k}(s_1)\right]_+ \leqslant \max\left\{\mathcal{O}\left(\frac{|\mathcal{A}|\bar{b}d^2\sqrt{K}}{\theta(1-\gamma)^2}\right) - K\zeta, 0\right\}.$$

Thus, the result follows. $\qquad\square$

# E   AUXILIARY LEMMAS

First, we present the MLE guarantee. Regarding the proof, refer to (20, Theorem 21). Note $\hat{P}_k$ and $\bar{\pi}_k$ are the quantities appearing in the proposed algorithms. We can also immediately obtain the statement to the offline case.

**Lemma 27** (MLE guarantee). *For a fixed episode $k$, with probability $1 - \delta$,*

$$\mathbb{E}_{s\sim\{0.5\rho_k+0.5\rho_k'\},a\sim U(\mathcal{A})}[\|\hat{P}_k(\cdot\,|\,s,a) - P^\star(\cdot\,|\,s,a)\|_1^2] \lesssim \zeta,$$

*where $\zeta := \frac{\ln(|\mathcal{M}|/\delta)}{k}$. As a straightforward corollary, with probability $1 - \delta$, $\forall k \in \mathbb{K}^+$,*

$$\mathbb{E}_{s\sim\{0.5\rho_k+0.5\rho_k'\},a\sim U(\mathcal{A})}[\|\hat{P}_k(\cdot\,|\,s,a) - P^\star(\cdot\,|\,s,a)\|_1^2] \lesssim 0.5\zeta_k, \qquad (27)$$

*where $\zeta_k := \frac{\ln(|\mathcal{M}|k/\delta)}{k}$.*

The following is a standard inequality to prove regret bounds for linear models. Refer to (20, Lemma G.2.)

**Lemma 28.** *Consider the following process. For $k = 1, \cdots, K$, $M_k = M_{k-1} + G_k$ with $M_0 = \lambda_0 I$ and $G_k$ being a positive semidefinite matrix with eigenvalues upper-bounded by $1$. We have that:*

$$2\ln\det(M_K) - 2\ln\det(\lambda_0 I) \geqslant \sum_{k=1}^{K} Tr(G_k M_{k-1}^{-1}).$$

**Lemma 29** (Potential function lemma). *Suppose $Tr(G_k) \leqslant B^2$.*

$$2\ln\det(M_K) - 2\ln\det(\lambda_0 I) \leqslant d\ln\left(1 + \frac{KB^2}{d\lambda_0}\right).$$

**Lemma 30** (Simulation lemma). *Given two CMDPs $(P', r + b, g)$ and $(P, r, g)$, for any policy $\pi$, we have:*

$$V_{P',r+b}^\pi - V_{P,r}^\pi = \frac{1}{1-\gamma}\mathbb{E}_{(s,a)\sim d_{P'}^\pi}[b(s,a)$$
$$+ \gamma\mathbb{E}_{P'(s'|s,a)}[Q_{P,r}^\pi(s',\pi)] - \gamma\mathbb{E}_{P(s'|s,a)}[Q_{P,r}^\pi(s',\pi)]]$$

*and*

$$V_{P',r+b}^{\pi} - V_{P,r}^{\pi} = \frac{1}{1-\gamma}\mathbb{E}_{(s,a)\sim d_P^{\pi}}[b(s,a)+$$

$$\gamma\mathbb{E}_{P'(s'|s,a)}[Q_{P,r+b}^{\pi}(s',\pi)] - \gamma\mathbb{E}_{P(s'|s,a)}[Q_{P',r+b}^{\pi}(s',\pi)]].$$

*Similarly, give two CMDPs $(P',r,g+b)$ and $(P,r,g)$, for any policy $\pi$, we have:*

$$V_{P',g+b}^{\pi} - V_{P,g}^{\pi} = \frac{1}{1-\gamma}\mathbb{E}_{(s,a)\sim d_{P'}^{\pi}}[b(s,a)$$

$$+ \gamma\mathbb{E}_{P'(s'|s,a)}[Q_{P,g}^{\pi}(s',\pi)] - \gamma\mathbb{E}_{P(s'|s,a)}[Q_{P,g}^{\pi}(s',\pi)]]$$

*and*

$$V_{P',g+b}^{\pi} - V_{P,g}^{\pi} = \frac{1}{1-\gamma}\mathbb{E}_{(s,a)\sim d_P^{\pi}}[b(s,a)+$$

$$\gamma\mathbb{E}_{P'(s'|s,a)}[Q_{P,g+b}^{\pi}(s',\pi)] - \gamma\mathbb{E}_{P(s'|s,a)}[Q_{P',g+b}^{\pi}(s',\pi)]].$$

We have used the following result from the optimization which is proved in Lemma 9 in (19).

**Lemma 31.** *Let $Y^*$ be the optimal dual variable, and $C \geqslant 2Y^*$, then, if*

$$V_r^{\pi^*}(s_1) - V_r^{\tilde{\pi}}(s_1) + C[b - V_g^{\tilde{\pi}}(s_1)]_+ \leqslant \delta \tag{28}$$

*then*

$$[b - V_g^{\tilde{\pi}}(s_1)]_+ \leqslant \frac{2\delta}{C}. \tag{29}$$

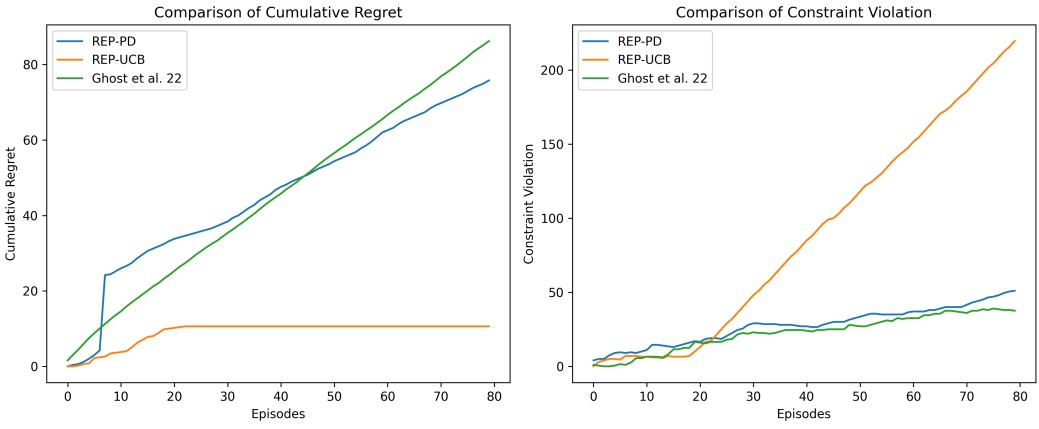

Figure 1: Comparison of regret and violation between REP-PD and REP-UCB.

## F  EXPERIMENT

To validate the convergence of REP-PD in CMDPs, we conduct experiments on a simulated job scheduling environment. Our implementation demonstrates the algorithm's ability to jointly learn latent representations while ensuring constraint satisfaction, without prior knowledge of transition dynamics.

The experimental setup is similar to (15; 16). We design a constrained MDP with discrete state space $\mathcal{S} = \{0, 1, \ldots, 9\}$, where $s$ represents the number of pending jobs. The action space $\mathcal{A} = \{0, 1\}$ allows the agent to either withhold jobs (a=0) or submit them (a=1). Total time horizon ($H$) is divided in 10 steps. The transition dynamics follow:

$$s_{h+1} = \begin{cases} \max\{s_h - 2a_h, 0\} & \text{with probability } 0.8 \\ \max\{s_h - a_h, 0\} & \text{with probability } 0.1 \\ s_h & \text{otherwise} \end{cases}$$

We assume that at time steps from 3 to 6, the reward is $1 - 0.9a$, In other time steps, the reward is $1 - 0.2a$. This mimics the setup where at certain time, it might be more costly to process a job. The utility function $g(s, a, s') = \frac{s-s'}{2}$ quantifies job processing efficiency, with a per-episode constraint $b - \sum_{h=0}^{H-1} g(s_h, a_h, s_{h+1}) \geqslant b = 2$, ensuring minimal job backlog.

We implement REP-PD with latent dimension $d = 2$, regularization $\lambda = 1$, exploration bonus coefficient $\alpha = 0.1$, and dual step size $\eta = 0.2$.

To demonstrate the effectiveness of our constraint-aware design, we compare against REP-UCB (21), an unconstrained low-rank MDP approach using similar representation learning but without dual variable updates, and a linear method (15) where the transition feature is known. As shown in Figure 1, REP-PD achieves comparable regret to REP-UCB while significantly reducing cumulative constraint violations. Notably, the performance of REP-PD is similarly to that of the linear method. This observation aligns with our theoretical analysis, which shows that in environments with small action spaces, the performance of REP-PD is similarly to that of the linear method. This validates that REP-PD successfully adapts representation learning techniques to the CMDP setting through the primal-dual architecture.

