# OpenReview forum: "Provably Safe Representation Learning in CMDPs: A Primal-Dual Approach"
_ICLR.cc/2026/Conference — Submitted to ICLR 2026_

### Official Review · Reviewer_yRUS · 2025-10-17

**Soundness:** 4
**Presentation:** 4
**Contribution:** 4
**Rating:** 6
**Confidence:** 4

**Summary:**

This paper studies the problem of representation learning in low-rank Constrained Markov Decision Processes (CMDPs), where the transition dynamics are unknown. The work focuses on developing principled algorithms that can efficiently learn in structured environments while adhering to safety or constraint requirements.

The authors propose REP-PD, the first provably efficient algorithm for low-rank CMDPs that simultaneously learns latent representations and optimizes policies. A key strength of the approach is that it achieves this while ensuring bounded constraint violations under the soft-constraint formulation, providing both theoretical guarantees and conceptual clarity on how representation learning interacts with constrained optimization in RL.

Furthermore, the paper extends its analysis to address both soft and hard constraint violation settings, offering a comprehensive treatment of constrained learning under the low-rank assumption. This dual focus enhances the generality and practical relevance of the proposed framework, positioning it as a meaningful step forward in the study of safe and structured reinforcement learning.

**Strengths:**

The paper is well-written and highly readable, with a clear logical flow throughout. Both the presentation of the main ideas and the structure of the proofs are easy to follow, making the technical content accessible and well-motivated.

The results are effectively highlighted and concisely summarized in the accompanying table, which greatly helps readers grasp the broader context and compare the performance and theoretical bounds across different methods. This clarity of presentation substantially enhances the paper’s readability and impact.

Moreover, the paper addresses an important and timely research direction, representation learning in safe reinforcement learning, which is a critical step toward building more efficient and reliable RL systems. Investigating how representation learning principles can be integrated into safety-constrained RL frameworks is both theoretically meaningful and practically significant.

**Weaknesses:**

It would greatly enhance the paper to include empirical results that complement the theoretical analysis. Even a small-scale or illustrative experiment could help demonstrate how the proposed method behaves in practice and validate the theoretical trends or assumptions. Empirical evidence would also provide readers with a clearer sense of the method’s applicability and robustness under realistic conditions.

In addition, it would be valuable to include an investigation of possible lower bounds to assess the tightness of the presented upper bounds. Establishing or discussing such lower bounds would complete the theoretical picture, helping to clarify whether the proposed rates are near-optimal or if there remains room for improvement.

**Questions:**

It would strengthen the theoretical contribution if the paper could include or discuss lower bound results to demonstrate the tightness of the presented upper bounds. Such results would help clarify whether the current rates are minimax-optimal and would provide a more complete understanding of the overall theoretical landscape.

Could you provide more intuition behind the K 3/4 convergence rate obtained in the hard-constrained safe RL setting? In particular, what factors lead to this exponent.

Do you believe that the rate could potentially be improved to K 1/2 under stronger assumptions or refined analysis? A brief discussion of the main bottlenecks limiting this improvement would also be insightful.

The paper appears to rely on the availability of an MLE oracle, which may not always be accessible in practical implementations. Could you elaborate on how the algorithm could be designed or approximated in the absence of such an oracle, and what impact this would have on the theoretical guarantees and computational efficiency?

It would be helpful if the authors could comment on the computational complexity of the proposed algorithms, especially regarding the scaling with respect to the rank, feature dimension, and number of samples. A comparison with existing approaches would further clarify the practical feasibility of the method.

Have you considered extending the framework to a model-free setting, where only the feature class Phi or density class mu (reference: Reinforcement Learning in Low-Rank MDPs with Density Features) is available?

Such an extension and discussion on this connection could help position the current work better within the broader literature on representation learning in low-rank RL.

**Details Of Ethics Concerns:**

nan

---

> ### Author Response · Authors · 2025-11-17
> **Response to Reviewer yRUS**
>
> We appreciate the reviewer's constructive feedback. Your comments are highly valuable and enable us to further clarify the contributions and limitations of our current work.
>
> [Empirical Evidence]
>
> We agree that empirical evidence is essential for validating theoretical findings. To address this, we have included empirical results in Appendix F of the revised manuscript.
> The experiment, conducted on a simulated job scheduling environment, illustrates the practical behavior of our REP-PD algorithm in CMDPs. They specifically validate the convergence of REP-PD while demonstrating its ability to jointly learn latent representations and maintain constraint satisfaction without relying on prior knowledge of the transition dynamics, thus supporting our theoretical claims.
>
> [Lower Bounds and Tightness]
>
> The establishment of minimax lower bounds is a challenging theoretical task that often requires dedicated analysis. While establishing whether the derived upper bounds are near-optimal is an important component of a complete theoretical picture, to the best of our knowledge, the theory of lower bounds for CMDPs is still largely underdeveloped.
> We consider a rigorous treatment of lower bounds to be beyond the scope of the current paper and designate it as a promising direction for future work.
>
> [Intuition Behind the $K^{3/4}$ Convergence Rate]
>
> The $K^{3/4}$ rate for the hard constraint violation setting (REP-PD-Hard) for bounding constraint violation stems directly from the requirement to guarantee constraint satisfaction in every episode.
> Achieving this strong guarantee necessitates a careful, iterative optimization of the dual variable within the primal-dual framework. This optimization step, which is required to approximate the "perfect" dual variable and balance exploration with safety, introduces an inherent overhead factor of $O(K^{1/4})$ compared to the $O(K^{1/2})$ regret of REP-PD. This behavior is consistent with prior work, where methods addressing hard constraint violation in Linear CMDPs generally exhibit a larger bound compared to those addressing soft constraint violation [2,3].
>
> Improving this rate to $O(K^{1/2})$ may require stronger assumptions on the environment or the use of more complex averaging techniques, which we reserve for future investigation.
>
> [Reliance on MLE Oracle]
>
> We clarify that the MLE oracle is utilized primarily for theoretical simplicity and tractability in bounding the representation error.
> The MLE oracle can be replaced by other representation learning techniques (e.g., [1]) for learning the representation.
> Specifically, in the absence of an explicit MLE oracle, the representation can be practically learned using methods such as Spectral Decomposition Representation as proposed in [1], or other optimization-based approaches.
> As long as the chosen representation method can provide a theoretical guarantee on the representation error (similar to the bound established in our Lemma 26), the overall regret and constraint violation analysis of REP-PD and REP-PD-hard remains valid. The key is the quality of the learned features, not the specific learning method.
>
> [Computational Complexity]
>
> To the best of our knowledge, our work provides the first representation learning method for reward-known CMDPs. The computational complexity, assuming access to the MLE oracle, is comparable to methods for linear CMDPs. Specifically, denoting $n_k$ as the number of samples in episode $k$, the computation required by lines 7, 8, and 9  of REP-PD incurs a cost of
> $O(d^2n_k)$,$O(d^3)$,$O(|\mathcal{A}|)$, respectively.
> The overall complexity per episode is thus polynomial in the feature dimension $d$ and the action space size $|\mathcal{A}|$, making it computationally feasible. In addition, REP-PD-hard also remains polynomial in $d$ and $|\mathcal{A}|$. The complexity is of the same order as existing linear CMDP approaches such as [2], further indicating that the proposed method is practically feasible in settings where low-rank structure holds.
>
> [Extension to Model-Free Setting]
>
> Extending the framework to a purely model-free setting is a compelling and significant research direction. While the current paper focuses on the model-based approach, the overall regret and constraint violation analysis of REP-PD and REP-PD-hard remains valid if a model-free learner produces representations for which one can establish explicit bounds on the representation error.
> We agree that the model-free extension is an important next step, which we are actively exploring as future work.
>
> [1]T.Ren, T.Zhang, L.Lee, J.E.Gonzalez, D.Schuurmans, and B.Dai, Spectral decomposition representation for reinforcement learning.
>
> [2]A.Ghosh,X.Zhou, and N.Shroff, Provably efficient model-free constrained rl with linear function approximation.
>
> [3]A.Ghosh,X.Zhou, and N.Shroff, Towards achieving sub-linear regret and hard constraint violation in model-free rl.

---

> > ### Author Response · Authors · 2025-11-26
> > **Kindly Checking In as the Discussion Period Nears Its End**
> >
> > Dear Reviewer,
> >
> > I hope this message finds you well. As the discussion period is nearing its end with less than one week remaining, l wanted to ensure we have addressed all your concerns satisfactorily. If there are any additional points or feedback you'd like us to consider, please let us know.Your insights are invaluable to us, and we'e eager to address any remaining issues to improve our work.
> >
> > Thank you for your time and effort in reviewing our paper.

---

> ### Comment · Reviewer_yRUS · 2025-11-27
>
> Thanks the authors for the great response!
>
> This paper looks great. I keep my score for marginal accept.

---

### Official Review · Reviewer_kdsa · 2025-10-19

**Soundness:** 2
**Presentation:** 3
**Contribution:** 2
**Rating:** 4
**Confidence:** 4

**Summary:**

This work focuses on representation learning within low-rank Constrained MDPs (CMDPs) where the feature $\phi$ is initially unknown. The authors introduce two Primal-Dual algorithms, REP-PD and REP-PD-hard, to learn a policy that minimizes regret while providing theoretical guarantees for both soft and hard constraint satisfaction on the utility function.

**Strengths:**

1. This is the first work to study low-rank Constrained Markov Decision Processes (CMDPs), eliminating the typical requirement for prior knowledge of the feature mapping $\phi$ in standard linear MDPs.

2. The paper provides comprehensive theoretical guarantees for the regret while successfully addressing both soft and hard  utility function constraints.

**Weaknesses:**

1. The novelty of this work appears limited. As outlined in Section 3.1, the proposed algorithm relies on three primary components: Representation Learning, Uncertainty-Aware Exploration, and Constraint-Guided Policy Optimization with Primal-Dual Adaptation. The first two elements are standard techniques widely adopted in previous analyses of low-rank MDPs. Similarly, the primal-dual method is a well-studied approach in Constrained MDPs (CMDPs). Consequently, the paper seems to present an integration of existing methods from the low-rank MDP and CMDP literature to solve the constrained low-rank MDP problem, raising concerns about its fundamental novelty.

2. A significant weakness lies in the lack of clarity regarding the definition and usage of the reward and utility functions within Algorithm 1 and Algorithm 2.The paper highlights a challenging scenario (Line 177) where the agent only receives bandit feedback. This necessitates the construction of estimations for the unknown reward and utility functions directly from this sparse feedback. However, the presented algorithms appear to directly utilize the actual functions, despite these functions being neither known nor specified as inputs to the algorithms.

The authors must explicitly clarify how the unknown reward and utility functions are estimated or approximated from the bandit feedback and then integrated into the Primal-Dual framework. Furthermore, a discussion is required to assess whether the uncertainty and estimation errors introduced by the unknown nature of these functions pose additional technical challenges to the regret and violation analysis of low-rank CMDPs.

**Questions:**

1. The citations in lines 1423 and 1436 are unclear. What is the mean of "refer to (author?)"?

2. The claim in line 370—that the sum over episodes is bounded via the elliptical potential lemma—appears to be improper.The standard elliptical potential lemma requires a fixed feature prefix (e.g., $\Sigma_k =\phi(s,a)\phi(s,a)^{\top} + \Sigma_{k-1}$). However, in the context of low-rank MDPs, the estimated feature $\hat{\phi}_k$ changes over time, which directly affects the prefix feature and the corresponding covariance matrix. Therefore, the authors cannot directly invoke the standard elliptical potential lemma. Further explanation and derivation are necessary when dealing with time-varying estimated features.

---

> ### Author Response · Authors · 2025-11-17
> **Response to Reviewer kdsa**
>
> We thank the reviewer for the feedback. Below we provide a detailed response to each of your feedback and we hope our response could address your concerns.
>
> [On Novelty of This Work]
>
> Thank you for the thoughtful comments regarding the novelty of our work. Our goal is to develop a provably efficient method for low-rank CMDPs that simultaneously learns latent transition representations and optimizes policies under constraints. While the high-level components of our algorithm—representation learning, uncertainty-aware exploration, and primal-dual policy optimization—may appear standard when viewed individually, the central technical difficulty lies in how these components interact under representation error and constraints.
>
> As shown in the proofs of Theorem 19 and Theorem 23, once representation error is present, balancing exploration, reward maximization, and constraint satisfaction becomes highly nontrivial. The algorithm must perform safe and effective exploration guided solely by learned features, without access to the true model. This fundamentally changes the analysis: prior techniques used in linear CMDPs—particularly those relying on pointwise transition-model error bounds—can no longer be applied. The presence of latent low-rank structure forces us to develop a different proof strategy that accounts for how representation inaccuracies propagate into both regret and constraint violation.
> Concretely, our proof strategy proceeds by applying a simulation-lemma-style argument inside the learned model and carefully controlling how the bonus computed under the learned features corresponds to the elliptical potential under the ground-truth features (Lemma 15 -18). This allows us to tightly characterize how representation error influences both the optimistic value estimates and the constraint-related quantities, thereby ensuring valid regret and violation guarantees under imperfect representations.
>
> Thus, the main novelty of our work lies not in the superficial combination of existing modules, but in the theoretical framework that unifies low-rank representation learning with constrained policy optimization. We provide the first regret and constraint-violation guarantees for low-rank CMDPs, establishing new connections between representation error and constraint feasibility that are essential for the analysis but absent in prior works.
>
>
> [On the Use of Reward and Utility Functions under Bandit Feedback]
>
> We thank the reviewer for this helpful comment. Our intention was to state that the algorithm has access to the reward and utility feedback $r$ and $g$, but not to the transition model $P$. Actually, the only stage where the reward and utility information is used is in the final policy update, while representation learning and exploration do not require access to these functions. In many CMDP applications—such as stochastic online environments, real-world robotic control, and queueing systems—the reward and utility signals are observable, whereas the transition dynamics are not.
>
> However, we acknowledge that the current phrasing in the manuscript may unintentionally suggest that the agent only observes reward and utility values for the actions it executes, which is not accurate.  This imprecise phrasing in the manuscript is a presentation oversight, and we have corrected it in the revision to make the distinction explicit.
>
> [On the Use of the Elliptical Potential Lemma]
>
> We thank the reviewer for raising this technical concern. As shown in the proof (Lines 1065–1079), through a sequence of one-step back inequalities and related transformations, we ultimately need to bound $\|\phi^*\|$, which is indeed handled using the elliptical potential lemma.
> Regarding the use of time-varying estimated features, this case is properly addressed in our analysis. Specifically, the bound
>
> $\mathbb{E}_{s\sim \rho_k,a\sim  U(\mathcal{A})}$
> [$\|\hat \phi_k(s,a)\|^2$]
>
> =Tr($\mathbb{E}_{\rho_k\times U(\mathcal{A})}[\hat \phi_k\hat \phi^{\top}_k](k\mathbb{E}_{\rho_k\times U(\mathcal{A})}[\hat \phi_k\hat \phi^{\top}_k]+\lambda_k I)^{-1} $)$\leq d/k$.
>
> Here, the norm $\|\cdot\|$ is the Mahalanobis norm with respect to $\Sigma^{-1}_{\rho_k\times U(\mathcal{A}),\hat \phi_k}$.
>
> Thus the varying features can still be controlled.
>
> [On Citation Clarity]
>
> We appreciate the reviewer for noting this issue. The unclear references have been corrected in the revised manuscript.

---

> > ### Author Response · Authors · 2025-11-26
> > **Kindly Checking In as the Discussion Period Nears Its End**
> >
> > Dear Reviewer,
> >
> > I hope this message finds you well. As the discussion period is nearing its end with less than one week remaining, l wanted to ensure we have addressed all your concerns satisfactorily. If there are any additional points or feedback you'd like us to consider, please let us know.Your insights are invaluable to us, and we'e eager to address any remaining issues to improve our work.
> >
> > Thank you for your time and effort in reviewing our paper.

---

> > > ### Comment · Reviewer_kdsa · 2025-11-28
> > >
> > > Thanks to the authors for the response. I agree that the main difficulty comes from the estimation of the transition kernel, and assuming a known reward function is tolerable if this assumption is explicitly clarified.
> > >
> > > Regarding the novelty, there exists some novelty in the analysis, but the framework (representation learning, uncertainty-aware exploration, and primal-dual policy optimization) is standard. Therefore, this seems like a borderline paper, marginally below the acceptance threshold. However, I would not mind if the paper is accepted.

---

### Official Review · Reviewer_SCmB · 2025-11-01

[review text omitted: it was posted to a different submission]

---

> ### Author Response · Authors · 2025-12-01
> **Likely Misattributed to Our Submission**
>
> Thank you for your time and effort in reviewing our submission. We would like to respectfully note that the comment appears to be unrelated to the content of our paper. The comment discusses concepts and methods that do not appear in our submission, which suggests that it may have been inadvertently copied from or intended for another paper.
>
> We completely understand that such mix-ups can occur during a busy review cycle. If possible, we would greatly appreciate it if the reviewer could briefly verify whether the comment was meant for our submission.

---

### Official Review · Reviewer_Mp9D · 2025-11-01

**Soundness:** 2
**Presentation:** 3
**Contribution:** 2
**Rating:** 4
**Confidence:** 3

**Summary:**

This paper studies constrained markov decision process (CMDP) under low-rank assumption. It proposed two algorithms for learning CMDP under soft-constraint and hard-constraint separately. The paper also theoretically proved the regret and constraint violation.

**Strengths:**

1. The study of CMDP under low-rank assumption is relatively new and significant. In particular, the proposed algorithm for hard constraint use a principled primal-dual mechanism with adaptive Lagrange updates.

**Weaknesses:**

1. The definition and use of regret seems strange to me. The regret is supposed to be the accumulative sub-optimal gaps of deployed policy by the RL agent. However, in the algorithm, the actual policy deployed is a concatenation of $\pi_k$ and uniform exploration. Thus, there is a significant concern about the validity of the regret and also the constraint violation proved in the paper.

**Questions:**

Please see the weakness part.

---

> ### Author Response · Authors · 2025-11-17
> **Response to Reviewer Mp9D**
>
> We thank the reviewer for raising this concern. We would first like to clarify that in our definition of regret, the policy $\pi_k$ is exactly the deployed policy produced by the algorithm.
> The uniform exploration data required in the each iteration is more like the intrinsic property of the feedback oracle, rather than part of the deployed policy $\pi_k$. Specifically, in line 4 of REP-PD, we provide the deployed policy $\pi_k$ to the feedback oracle, which then returns samples generated from $\pi_k$ mixed with uniform samples.
> This view aligns closely with standard stochastic-gradient models in online optimization, where the oracle returns a random estimate of the true gradient.
> Note that, the mixed feedback oracle we adopt here is widely used throughout the RL literature to ensure sufficient exploration [7-9]. Our notions of regret and constraint violation follow the standard definitions used in RL and CMDP works [1–6].
>
> Therefore, all theoretical guarantees in the paper are established for the deployed policies $\pi_k$. The uniform exploration affects only the oracle’s feedback mechanism—not the definition of regret or constraint violation—and therefore does not compromise the validity of our performance guarantees.
>
>
> [1]A.Bura, A.HasanzadeZonuzy, D.Kalathil, S.Shakkottai, and J.-F.Chamberland, Dope: Doubly optimistic and pessimistic exploration for safe reinforcement learning.
>
> [2]H.Wei, X.Liu, and L.Ying, Safe reinforcement learning with instantaneous constraints: the role of aggressive exploration.
>
> [3]S.Amani, C.Thrampoulidis, and L.Yang, Safe reinforcement learning with linear function approximation.
>
> [4]A.Ghosh,X.Zhou, and N.Shroff, Provably efficient model-free constrained rl with linear function approximation.
>
> [5]A.Ghosh,X.Zhou, and N.Shroff, Towards achieving sub-linear regret and hard constraint violation in model-free rl.
>
> [6]T.Kitamura, A.Ghosh, T.Kozuno, W.Kumagai, K.Kasaura, K.Hoshino, Y.Hosoe, and Y.Matsuo, Provably efficient rl under episode-wise safety in linear cmdps.
>
> [7]T.Ren, T.Zhang, L.Lee, J.E.Gonzalez, D.Schuurmans, and B.Dai, Spectral decomposition representation for reinforcement learning.
>
> [8]A.Agarwal, S.Kakade, A.Krishnamurthy, and W.Sun, Flambe: Structural complexity and representation learning of low rank mdps.
>
> [9]C.Ni, Y.Song, X.Zhang,C.Jin, and M.Wang, Representation learning for general-sum low-rank markov games.

---

> > ### Author Response · Authors · 2025-11-26
> > **Kindly Checking In as the Discussion Period Nears Its End**
> >
> > Dear Reviewer,
> >
> > I hope this message finds you well. As the discussion period is nearing its end with less than one week remaining, l wanted to ensure we have addressed all your concerns satisfactorily. If there are any additional points or feedback you'd like us to consider, please let us know.Your insights are invaluable to us, and we'e eager to address any remaining issues to improve our work.
> >
> > Thank you for your time and effort in reviewing our paper.

---

> > ### Comment · Reviewer_Mp9D · 2025-11-28
> >
> > Thanks for the response. While the authors' response could be a situation of uniform action selection, it is not a satisfying reason.
> >
> > First, 7-9 use such mix exploration without concern since they focus on sample complexity rather than regret. For a regret minimization and constraint violation minimization problem, I believe any uniform exploration should be taken into account.
> >
> > Second, even in practical scenarios, where epsilon-greedy, softmax policy, or stochastic gradient is deployed, the randomness is controlled rather than pure uniform.
> >
> > One of possible improvement is to consider epsilon-greedy-style exploration but with decay. E.g. I am wondering what would happen if we use $\pi\_k$ with probability $1-\epsilon\_k$ and uniformly explore w.p. $\epsilon\_k$, and $\epsilon\_k$ deca y w.r.t. $k$. Perhaps this would ensure sublinear regret and controlled violation at the same time.
> >
> > In summary, my opinion to this work currently is neutral.

---

> > > ### Author Response · Authors · 2025-12-01
> > > **Response to Reviewer Mp9D**
> > >
> > > Thank you for the follow-up comments. We would like to clarify that the use of uniform action selection in our analysis is both valid and theoretically justified. Indeed, the mixed exploration scheme we adopt is standard in the literature [1-4], and, importantly, these works first establish regret bounds under such exploration strategy and then convert them into PAC-style sample-complexity guarantees. This is explicitly how [1–4] proceed: each of these papers derives a regret bound under a uniform exploration term (e.g., Lemma 13 in [1], Lemma 8 in [2], Lemma B.8 in [3], Lemma 9 in [4]) before turning the regret guarantee into sample complexity. Our analysis follows the same structure. In fact, our regret bound can also be directly converted to a sample-complexity guarantee, and we have added a standard regret-to-PAC conversion (Corollary 20 in the revision) to make this clear. Thus, incorporating a uniform exploration component is fully consistent with established regret analyses and does not affect our theoretical guarantees.
> > >
> > > From a practical viewpoint, uniform sampling is also a simple and widely used exploration mechanism—it introduces minimal implementation overhead and provides guaranteed coverage of all actions. We agree that more refined schemes such as decaying $\epsilon$-greedy strategy may lead to improved practical regret performance and better control of constraint violations. However, incorporating such scheme necessitates a careful, application-specific tuning of the decay schedule, which can be particularly complex in dynamic, constrained environments. We appreciate the reviewer for pointing out this potential extension, and we leave the exploration and validation of such advanced exploration strategies as an important direction for future work.
> > >
> > > [1]T. Ren, T. Zhang, L. Lee, J.E. Gonzalez, D. Schuurmans, and B. Dai, Spectral decomposition representation for reinforcement learning.
> > >
> > > [2]T. Ren,C. Xiao, T. Zhang, et al. Latent variable representation for reinforcement learning.
> > >
> > > [3]C. Ni, Y. Song, X. Zhang,C. Jin, and M. Wang, Representation learning for general-sum low-rank markov games.
> > >
> > > [4]M. Uehara, X. Zhang, and W. Sun, Representation learning for online and offline rl in low-rank mdps.

---

### Author Response · Authors · 2025-12-01
**Global Response**

We thank all reviewers for their thoughtful and constructive feedback. This paper tackles a fundamental challenge in constrained reinforcement learning: how to achieve provably safe and sample-efficient learning in CMDPs when the transition model is unknown and the state space is high-dimensional. Unlike prior tabular or linear CMDP methods, which either suffer from state-space dependence or require the true latent features to be known in advance, our work addresses the core difficulty of jointly learning latent representations and optimizing constrained policies—a task complicated by the need to balance exploration, reward maximization, and constraint satisfaction under imperfectly learned features. We develop the first provably efficient framework for low-rank CMDPs that integrates representation learning, optimistic value estimation, and primal–dual policy updates into a single analyzable pipeline. Our approach overcomes the key difficulty of performing safe exploration with learned features, and achieves regret and constraint-violation bounds that scale with the intrinsic low-rank dimension rather than the state space. The framework further supports variants guaranteeing zero soft violation or handling hard constraints with near-optimal bounds. We appreciate that multiple reviewers recognized the significance of establishing such a general theoretical foundation. Below we address the key concerns raised across reviews.

[Mixed Exploration and  Regret]
We appreciate reviewer Mp9D’s comments on the use of uniform exploration and its interaction with regret and constraint violation. In our setting, the deployed policy in each episode is the learned policy $\pi_k$, while the uniform component comes from the feedback oracle, which mixes uniform samples into the data. This mechanism aligns with standard stochastic oracles widely adopted in low-rank and linear MDP works, where regret bounds are first derived under mixed exploration and later converted to PAC guarantees. To further emphasize this connection, we added a regret-to-PAC conversion in the revision.
More refined schemes, such as decaying $\epsilon$-greedy, may improve empirical performance, and we highlight this as an interesting direction for future work. Our theoretical guarantees, however, remain valid under the mixed-feedback oracle and follow established precedents in the literature.

[Novelty and Contributions]
While the high-level components—representation learning, uncertainty-aware exploration, and primal-dual policy optimization—may resemble elements from prior works, the main novelty of our work lies in the integration of these components under low-rank structure and constraints, and in the new analytical framework required to handle representation error in CMDPs. Unlike linear CMDPs, where exact features are known, the low-rank setting requires learning latent transition representations, and these learned features introduce nontrivial coupling between exploration, value estimation, and constraint feasibility. Our analysis develops a simulation-lemma-style argument within the learned model and carefully controls how representation error propagates into both regret and constraint violation bounds. This yields the first regret and violation guarantees for low-rank CMDPs and establishes novel conceptual connections between representation error, safety, and constrained policy optimization.

[Elliptical Potential Lemma With Time-Varying Features]
We handle time-varying estimated features by bounding their Mahalanobis norm under the evolving covariance matrix and using expectations over the uniform exploration distribution, yielding a valid elliptical-potential-style bound. The revision makes this explicit.

[Empirical Validation]
Following reviewer yRUS’s suggestion, Appendix F includes an experiment on a simulated job scheduling environment, showing that our algorithm can learn latent representations and satisfy constraints without prior knowledge of transitions, validating theoretical predictions.

[MLE Oracle and Practical Implementability]
The MLE oracle is used in our analysis primarily for clarity and to bound the representation error. In practice, it can be replaced with other representation-learning procedures, such as spectral decomposition or optimization-based methods, provided that they deliver comparable error guarantees. Our results rely on the quality of the learned features rather than on any specific oracle.

[Computational Complexity and Model-Free Extensions]
The algorithm’s cost is polynomial in feature and action dimensions, matching existing linear CMDP methods. Extending to model-free settings is possible if the learner guarantees bounded representation error.

We thank all reviewers once again for their insightful comments. Their suggestions have helped us significantly strengthen both the theoretical and empirical presentation of our work.

---

### Meta-Review · Area_Chair_ePDc · 2026-01-09

**Summary:**

This paper studies representation learning in low-rank CMDP with unknown dynamics. The authors propose REP-PD, an algorithm that integrates representation learning (via MLE) with a primal-dual policy optimization approach. The paper provides theoretical guarantees for both soft and hard constraint violations with sample complexity independent of the state space dimension. A variant, REP-PD-hard, is introduced to handle strict constraint satisfaction. Discarding one of the reviewers' comments, the rest put the paper on the borderline. They concerned about the experimental evaluation, novelty, and technical correctness. After the rebuttal, reviewers tend to maintain their scores. The AC tends to agree with the reviewers' concern about novelty and hence recommends weak rejection.

**Reviewer Concerns:**

- Reviewer yRUS (Score: 6) was positive, praising the paper as the first provably efficient algorithm for this setting. They requested empirical validation, which the authors provided in the revision (Appendix F).

- Reviewer kdsa (Score: 4) initially criticized the novelty, viewing the work as a combination of standard components. After the rebuttal, they acknowledged that the analysis of error propagation in this specific setting holds some novelty. But he is not convinced that the paper  is above the borderline.

- Reviewer Mp9D (Score: 4) raised technical concerns regarding the definition of regret under mixed exploration (uniform sampling). The reviewer remained neutral/borderline after the rebuttal.

- AC disregards Reviewer SCmB's feedback entirely as it applies to a different paper.

**Reviewer Scores:**

See Reviewer's concerns.

---

### Decision · Program_Chairs · 2026-01-26

Reject